# The E3 ubiquitin ligase component, Cereblon, is an evolutionarily conserved regulator of Wnt signaling

Chen Shen[1,2,9], Anmada Nayak[1,9], Leif R. Neitzel[3], Amber A. Adams[4], Maya Silver-Isenstadt[3], Leah M. Sawyer[5], Hassina Benchabane[4], Huilan Wang[1], Nawat Bunnag[4], Bin Li[1], Daniel T. Wynn[1], Fan Yang[1,2], Marta Garcia-Contreras [1], Charles H. Williams[3], Sivanesan Dakshanamurthy[6], Charles C. Hong [3], Nagi G. Ayad[6,7,8], Anthony J. Capobianco [1,8], Yashi Ahmed[4], Ethan Lee[5] & David J. Robbins [1,6,8 ✉]

Immunomodulatory drugs (IMiDs) are important for the treatment of multiple myeloma and myelodysplastic syndrome. Binding of IMiDs to Cereblon (CRBN), the substrate receptor of the CRL4$^{CRBN}$ E3 ubiquitin ligase, induces cancer cell death by targeting key neo-substrates for degradation. Despite this clinical significance, the physiological regulation of CRBN remains largely unknown. Herein we demonstrate that Wnt, the extracellular ligand of an essential signal transduction pathway, promotes the CRBN-dependent degradation of a subset of proteins. These substrates include Casein kinase 1α (CK1α), a negative regulator of Wnt signaling that functions as a key component of the β-Catenin destruction complex. Wnt stimulation induces the interaction of CRBN with CK1α and its resultant ubiquitination, and in contrast with previous reports does so in the absence of an IMiD. Mechanistically, the destruction complex is critical in maintaining CK1α stability in the absence of Wnt, and in recruiting CRBN to target CK1α for degradation in response to Wnt. CRBN is required for physiological Wnt signaling, as modulation of *CRBN* in zebrafish and Drosophila yields Wnt-driven phenotypes. These studies demonstrate an IMiD-independent, Wnt-driven mechanism of CRBN regulation and provide a means of controlling Wnt pathway activity by CRBN, with relevance for development and disease.

[1] Molecular Oncology Program, The DeWitt Daughtry Family Department of Surgery, Miller School of Medicine, University of Miami, Miami, FL, USA. [2] The Sheila and David Fuente Graduate Program in Cancer Biology, Miller School of Medicine, University of Miami, Miami, FL, USA. [3] Department of Medicine, University of Maryland, Baltimore, MD, USA. [4] Department of Molecular and Systems Biology and the Norris Cotton Cancer Center, Geisel School of Medicine, Dartmouth College, Hanover, NH, USA. [5] Department of Cell and Developmental Biology, Vanderbilt University, Nashville, TN, USA. [6] Department of Oncology, Lombardi Comprehensive Cancer Center, Georgetown University, Washington, DC, USA. [7] Center for Therapeutic Innovation, Department of Neurological Surgery, Miami Project to Cure Paralysis, Miller School of Medicine, University of Miami, Miami, FL, USA. [8] Sylvester Comprehensive Cancer Center, Miller School of Medicine, University of Miami, Miami, FL, USA. [9] These authors contributed equally: Chen Shen, Anmada Nayak. ✉email: dr956@georgetown.edu

The Wnt signaling pathway is one of the major signal transduction cascades that regulate metazoan development and control tissue homeostasis. Consistent with these essential physiological roles, deregulated Wnt signaling is a driver of many human disease states[1]. In the absence of Wnt ligands, the pivotal transcriptional coactivator β-Catenin is targeted for proteasomal degradation via its association with a macromolecular protein complex termed the destruction complex[2–6]. This complex includes the Wnt regulators Adenomatous polyposis coli (APC)[3], Axin[4,5], Glycogen synthase kinase 3 (GSK3)[4,6], and Casein kinase 1α (CK1α)[5,6]. Upon Wnt stimulation, the degradation of β-Catenin via the destruction complex is attenuated[7,8], allowing β-Catenin to accumulate in the nucleus and regulate the transcription of specific target genes[9,10]. Consistent with its essential role in the destruction complex, CK1α has been identified as one of the crucial negative regulators of Wnt signaling[11,12]. Although we and others have reported that CK1α is important to this process during metazoan development and in disease settings[13–16], little is known about how CK1α itself is regulated in cells.

A number of small molecules have been shown to control the activity of CK1α[14–21], amongst which the immunomodulatory drug (IMiD) lenalidomide induces CK1α protein degradation[18,21]. This process is mediated by a macromolecular E3 ubiquitin ligase complex consisting of the ligase scaffold Cullin-4 (CUL4), the RING-finger protein RING-box1 (RBX1), the adapter Damage-specific DNA binding protein 1 (DDB1), and the substrate receptor Cereblon (CRBN) (CRL4[CRBN])[18,21–23]. Thus, when bound to such IMiDs, CRBN is able to target a number of neosubstrates for degradation, including Ikaros and CK1α, thereby inhibiting cancer growth[18,24–31]. Despite this established role of CRBN as an anticancer drug target, its physiological role remains poorly understood. Here, we demonstrate a small molecule-independent, evolutionarily conserved mechanism in which Wnt pathway activation stimulates CRBN to associate with CK1α and mediate ubiquitin-dependent CK1α degradation. This mechanism further modulates Wnt signal transduction, with importance for development and disease.

## Results

**Wnt stimulation induces ubiquitin-dependent, proteasomal degradation of CK1α.** Based on our previous work[15], which showed reduced levels of CK1α in APC mutant cells, we postulated that Wnt signaling might regulate CK1α protein levels. To test this, we treated HEK293T (HEK) cells with recombinant Wnt3a for varying lengths of time or concentrations and analyzed CK1α levels in cell lysates. Wnt3a treatment reduced CK1α levels in both a time- and concentration-dependent manner, relative to the control protein α-Tubulin (Fig. 1a and Supplementary Fig. 1a). We also noted that recombinant Frizzled8 cysteine-rich domain (FZD8-CRD)[32], a Wnt pathway antagonist, blocked the decrease of CK1α levels in response to Wnt3a (Supplementary Fig. 1b), consistent with Wnt3a acting in a specific manner. By contrast, CK1α gene expression was unaffected by Wnt exposure, relative to a Wnt target gene control (Fig. 1b), suggesting that Wnt downregulates CK1α protein levels via a posttranscriptional mechanism. Furthermore, we blocked new protein synthesis in HEK cells using cycloheximide and observed a substantially decreased CK1α half-life in response to Wnt3a (Fig. 1c and Supplementary Fig. 1c). Consistent with Wnt regulating the stability of only a subset of CK1α, the nuclear pool of CK1α was unaffected by Wnt treatment while its corresponding cytoplasmic pool was simultaneously degraded (Supplementary Fig. 1e). Further, unlike the general pool of cytoplasmic CK1α, CK1α associated with the Phosphatase and tensin homolog (PTEN)[33]

was not degraded in response to Wnt (Supplementary Fig. 1f). Taken together, these results show that Wnt signaling decreases the steady-state levels of a cytoplasmic pool of CK1α by promoting its turnover.

To determine if CK1α levels were regulated via a proteasome- or lysosome/autophagosome-mediated manner we stimulated HEK cells with Wnt3a in the presence of a proteasome (MG132) or lysosome/autophagosome (bafilomycin A1 (BA)) inhibitor. MG132 treatment prevented Wnt3a-dependent CK1α degradation (Fig. 1d), while BA treatment did not (Fig. 1e), suggesting that Wnt-dependent CK1α degradation occurs via the proteasome. To determine if CK1α is ubiquitinated in response to Wnt exposure we treated HEK cells with Wnt3a and then immunoprecipitated CK1α from these cell lysates and immunoblotted for ubiquitin. Wnt3a treatment induced significant levels of CK1α ubiquitination (Fig. 1f). Similar results were obtained using a tandem ubiquitin-binding entity (TUBE) assay (Supplementary Fig. 1d). These results demonstrate that Wnt stimulation results in increased levels of CK1α ubiquitination and subsequent degradation via the proteasome.

**Wnt-induced degradation of CK1α requires CRBN, the substrate receptor of the CRL4[CRBN] E3 ubiquitin ligase complex.** We noted that a pan-inhibitor of the RING-finger family of E3 ubiquitin ligases, MLN4924, attenuated Wnt3a-induced CK1α degradation (Supplementary Fig. 2a). We, therefore, evaluated several candidate RING-finger E3 ubiquitin ligases previously implicated in CK1α or Wnt regulation[18,34], knocking down their expression in HEK cells and examining CK1α levels (Supplementary Fig. 2b). Interestingly, depletion of CRBN[35,36], the substrate receptor of the CRL4[CRBN] E3 ubiquitin ligase complex, attenuated Wnt-induced CK1α degradation (Fig. 2a and Supplementary Fig. 2b–d). Furthermore, knockdown of CRBN expression prevented the Wnt-induced decrease in CK1α half-life (Fig. 2b and Supplementary Fig. 2e–g), suggesting that the CRL4[CRBN] E3 ubiquitin ligase complex regulates Wnt-driven CK1α degradation.

To better understand how Wnt induces CRBN-dependent degradation of CK1α, we performed an in vitro binding assay between purified CRBN and recombinant CK1α (Fig. 2c). CRBN was only associated with recombinant CK1α when it was isolated from Wnt-treated cells (Fig. 2d). Notably, CK1α is known to interact with CRBN in a manner dependent on some IMiDs[21]. Thus, the Wnt-dependent association of CRBN with CK1α is to a point consistent with this previous report, suggesting that the two proteins may require an additional factor/modification to associate upon Wnt stimulation[21]. We next took advantage of these purified proteins to perform an in vitro CK1α ubiquitination assay, directly determining if Wnt signaling activates CRBN-dependent ubiquitination (Fig. 2c). We found that ubiquitination of recombinant CK1α was significantly enhanced when CRBN was isolated from Wnt-treated cells versus control cells (Fig. 2e). These results suggest that Wnt signaling induces CRBN to bind to and mediate the ubiquitination of CK1α.

**The β-Catenin destruction complex regulates CRBN-mediated CK1α degradation.** As CK1α is an essential component of the β-Catenin destruction complex[5,6], we next determined the kinetics of Wnt-driven CK1α degradation relative to that of other components in the complex. Wnt3a-induced CK1α degradation occurred prior to alterations in the levels of other destruction complex proteins (Fig. 3a and Supplementary Fig. 3a). Importantly, in contrast to CK1α, any Wnt-driven decreases in the levels of other destruction complex components occurred at a later time point (24 h) and were CRBN-independent (Fig. 3b).

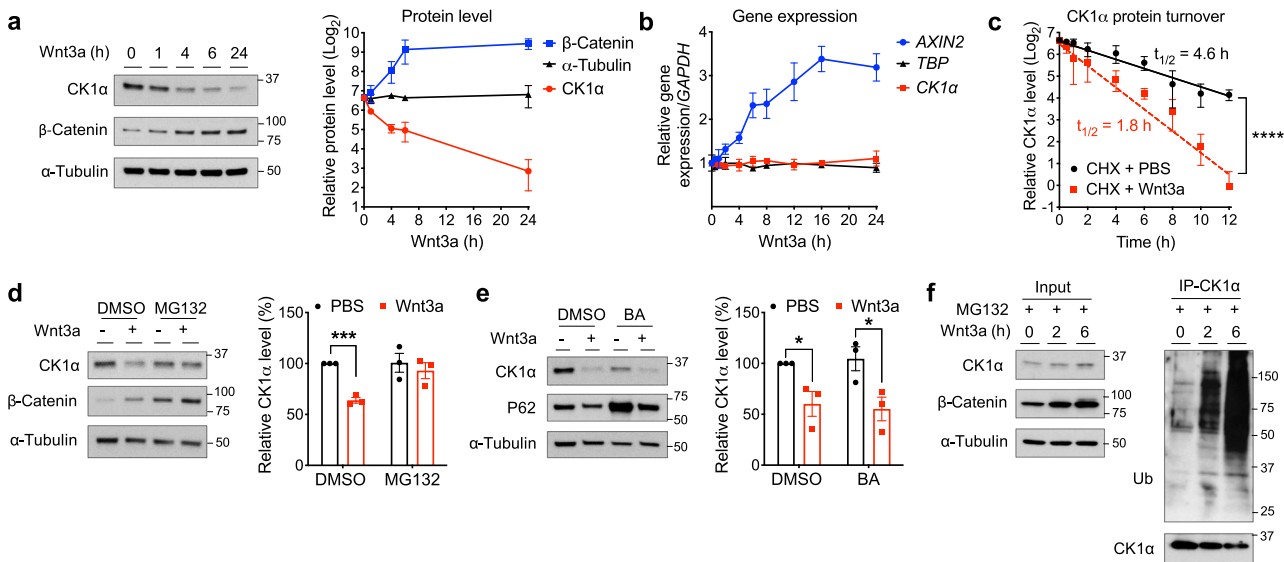

**Fig. 1 Wnt signaling regulates CK1α levels via ubiquitin-dependent proteasomal degradation. a** Extracts of HEK cells treated with Wnt3a for various lengths of time were evaluated by immunoblotting. A representative immunoblot (left panel) or quantitation of immunoblots (mean ± SEM, $n = 4$ independent experiments; right panel) is shown. **b** RNA from HEK cells treated with Wnt3a for various lengths of time was used to determine the expression of the indicated genes, using quantitative RT-PCR analysis. Quantification of gene expression (mean ± SEM, $n = 3$ independent experiments) is shown. **c** HEK cells were co-treated with cycloheximide and PBS or Wnt3a for the indicated time and extracts of these cells were evaluated by immunoblotting for CK1α or HSP90. CK1α levels from immunoblots were quantitated, normalized to that of HSP90, and plotted to determine CK1α turnover in response to Wnt3a (mean ± SEM, $n = 3$ independent experiments). Asterisks indicate statistical significance (two-way ANOVA analysis, ****$p$ value < 0.0001). **d** Extracts of HEK cells treated for 6 h with PBS or Wnt3a, together with DMSO or 10 μM MG132, were evaluated by immunoblotting. A representative immunoblot (left panel) and a quantification of immunoblots from (mean ± SEM, $n = 3$ independent experiments; right panel) are shown. **e** Extracts of HEK cells treated for 24 h with PBS or Wnt3a, together with DMSO or 20 nM bafilomycin A1 (BA), were evaluated by immunoblotting. A representative immunoblot (left panel) and a quantification of immunoblots (mean ± SEM, $n = 3$ independent experiments; right panel) are shown. P62 is a biomarker for lysosomal/autophagosomal inhibition. Asterisks in **d** and **e** indicate statistical significance (two-tailed Student's $t$-test, *$p$ value < 0.05, ***$p$ value < 0.001). **f** CK1α was immunoprecipitated from the lysates of HEK cells treated with Wnt3a and MG132 for various time points, followed by analyses of the indicated proteins by immunoblotting. A representative immunoblot ($n = 3$ independent experiments) is shown.

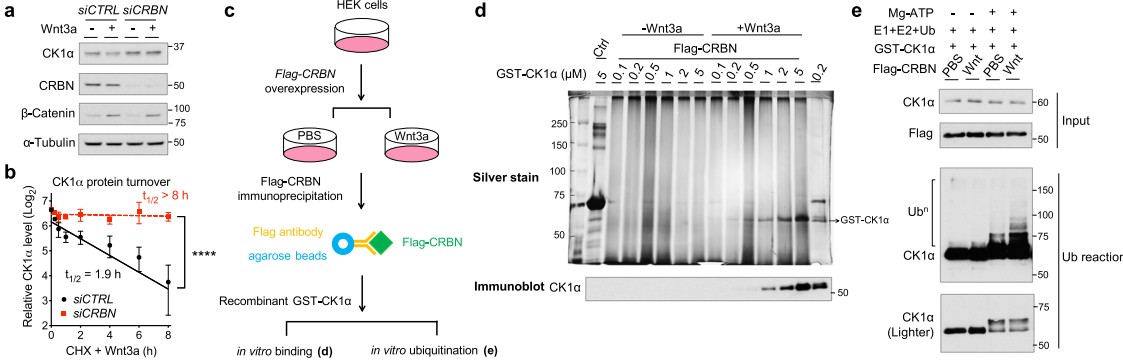

**Fig. 2 Wnt-induced degradation of CK1α requires CRBN, the substrate receptor of the CRL4^CRBN E3 ubiquitin ligase complex. a** HEK cells were transfected with the indicated smart-pool siRNA and subsequently treated with PBS or Wnt3a for 24 h. Extracts of these cells were evaluated by immunoblotting. A representative immunoblot ($n = 5$ independent experiments) is shown. **b** HEK cells were transfected with the indicated smart-pool siRNA and co-treated with Wnt3a and cycloheximide for the indicated time. Extracts of these cells were evaluated by immunoblotting. CK1α levels from immunoblots were quantitated and normalized to HSP90 levels (mean ± SEM, $n = 3$ independent experiments). Asterisks indicate statistical significance (two-way ANOVA analysis, ****$p$ value < 0.0001). **c** A schematic outlining experimental details of the in vitro binding and ubiquitination assays shown respectively in panels **d** and **e**. **d** HEK cells were transfected with a plasmid encoding Flag-CRBN and subsequently treated with PBS or Wnt3a for 4 h. Flag-CRBN immunoprecipitated from extracts of these cells were incubated with the indicated amounts of recombinant GST-CK1α at 4 °C for 1 h. Flag-CRBN beads were re-isolated, washed, and CK1α bound was eluted using sample buffer. These immunoprecipitates were subjected to SDS-PAGE, followed by silver staining (top panel) or immunoblotting (bottom panel). IgG beads serve as a control for Flag-CRBN beads (left lane) and recombinant GST-CK1α a loading control (right lane). A representative gel image and immunoblot ($n = 3$ independent experiments) is shown. **e** HEK cells were transfected with a plasmid encoding Flag-CRBN and subsequently treated with PBS or Wnt3a for 4 h. Flag-CRBN immunoprecipitated from extracts of these cells were incubated with a mixture of recombinant GST-CK1α, ubiquitin, and a mixture of E1 and E2 ligases in the presence of vehicle or Mg-ATP, and incubated at 30 °C for 3 h. These reaction mixtures were subsequently analyzed by immunoblotting for the indicated proteins. A representative immunoblot ($n = 3$ independent experiments) is shown.

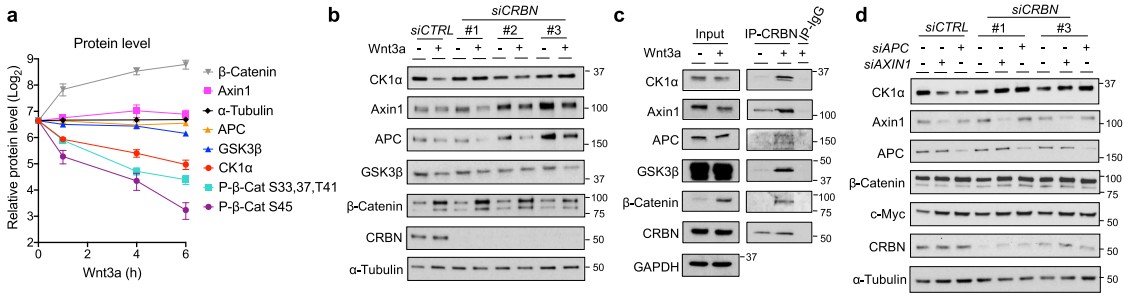

**Fig. 3 The β-Catenin destruction complex regulates CRBN-mediated CK1α degradation. a** Extracts of HEK cells treated with Wnt3a for various lengths of time were evaluated by immunoblotting. Quantitation of immunoblots (mean ± SEM, $n = 3$ independent experiments) is shown. Levels of phosphorylated β-Catenin were normalized to that of total β-Catenin. **b** HEK cells were transfected with *control* (*CTRL*) siRNA or one of three distinct *CRBN* siRNA, followed by PBS or Wnt3a treatment for 24 h. Extracts of these cells were evaluated by immunoblotting. **c** CRBN was immunoprecipitated from extracts of HEK cells treated with PBS or Wnt3a in the presence of MG132 for 4 h, followed by immunoblotting of the indicated proteins. **d** HEK cells were co-transfected with *control* (*CTRL*) siRNA or smart-pool *AXIN1* or *APC* siRNA, along with one of two distinct *CRBN* siRNA. Extracts of these cells were evaluated by immunoblotting. A representative immunoblot ($n = 3$ independent experiments) is shown in **b**–**d**.

These results suggest that Wnt-driven, CRBN-dependent CK1α degradation occurs via an intact, functional destruction complex. Based on this finding, we speculated that CRBN is recruited to the destruction complex in response to Wnt stimulation in order to target CK1α. To test this hypothesis, we immunoprecipitated endogenous CRBN from lysates of HEK cells, treated with or without Wnt3a, and probed these immunoprecipitates for components of the destruction complex. We found that CRBN did indeed associate with CK1α and other components of the destruction complex in response to Wnt3a, including the two pivotal scaffolding proteins Axin1 and APC (Fig. 3c). This result suggests that the destruction complex facilitates CRBN-dependent CK1α proteolysis upon Wnt stimulation.

Consistent with the destruction complex stabilizing CK1α in the absence of Wnt stimulation, knockdown of *AXIN1 or APC* expression induced CK1α degradation (Fig. 3d). Importantly, CK1α degradation following *AXIN1* or *APC* depletion was also CRBN-dependent (Fig. 3d). In cells deficient for *AXIN1* or *APC*, CK1α stabilization resulting from *CRBN* knockdown did not attenuate β-Catenin stabilization or downstream Wnt activity biomarkers (c-Myc). This observation is consistent with previous work demonstrating the importance of these proteins in efficiently targeting CK1α to β-Catenin[5,37]. Together, these results suggest that the integrity of the β-Catenin destruction complex is important for the regulation of CRBN-mediated CK1α proteolysis; the destruction complex both stabilizes CK1α in the absence of Wnt and promotes an activation step (via CRBN recruitment) upon Wnt induction.

**Wnt stimulation modulates the interaction of CRBN with CK1α in a manner that requires the IMiD binding pocket.** CRBN was reported previously to interact with and mediate CK1α ubiquitination only when the IMiD, lenalidomide, formed a molecular bridge between them[18,21]. This lenalidomide-induced, CRBN-dependent degradation of CK1α occurred via a β-hairpin loop anchored by Gly40[21] (Fig. 4a, residues labeled magenta with white text). Similarly, using mutagenesis of key residues, we found that this β-hairpin loop of CK1α is also important for its lenalidomide-independent, Wnt-driven proteolysis (Fig. 4b). We therefore further investigated the role of the IMiD binding pocket of CRBN (Fig. 4a, binding pocket in blue and pivotal CRBN residues labeled green with white text) in Wnt-induced CK1α proteolysis. We mutated five pivotal CRBN residues within this pocket, all of which significantly attenuated their ability to degrade and bind to CK1α (Fig. 4c, d). Thus, Wnt signaling induces CRBN-dependent CK1α degradation in a

manner that requires its previously described IMiD binding pocket[21].

Due to primary sequence differences between human and mouse CRBN, lenalidomide is not able to induce CK1α degradation in mouse cells[18]. Thus, we determined whether Wnt-induced CK1α degradation also exhibits this specificity. We compared Wnt3a- and lenalidomide-induced degradation of CK1α in mouse fibroblasts (NIH3T3 cells) and found that Wnt3a-induced CK1α degradation and its binding to CRBN in these mouse cells, whereas lenalidomide did not (Supplementary Fig. 4a, b). This is consistent with a conserved function of Wnt signaling in CRBN-mediated CK1α proteolysis, unlike that of lenalidomide.

**Wnt signaling induces CRBN-mediated degradation of a broad spectrum of endogenous substrates.** We next determined the ability of Wnt3a to induce the degradation of a broader subset of known, endogenous CRBN substrates, including Glutamine synthetase (GS)[38], c-Jun[39], and Meis homeobox 2 (MEIS2)[40]. We noted that in addition to causing decreased levels of CK1α, Wnt3a treatment also resulted in a decrease of the protein levels of these other CRBN substrates (Fig. 5a and Supplementary Fig. 5a), without affecting their levels of gene expression (Supplementary Fig. 5b–e). Similar to CK1α, the half-life of GS, c-Jun, and MEIS2 were also significantly decreased in response to Wnt3a, and this reduction was CRBN-dependent (Fig. 5b–g and Supplementary Fig. 5f–g). In contrast to CK1α, lenalidomide treatment had no significant effect on the degradation of these other CRBN substrates (Fig. 5h). Collectively, these results show that compared to lenalidomide, Wnt signaling activates the function of CRBN towards a wide spectrum of substrates.

**CRBN is a positive regulator of Wnt activity.** We used a reporter cell line stably expressing a TOPFlash Wnt reporter gene (HEK293STF) in order to begin to probe the role CRBN may play in Wnt pathway activity. We found that *CRBN* knockdown significantly attenuated Wnt-stimulated reporter gene activity (Fig. 6a and Supplementary Fig. 6a). Additionally, overexpression of *CRBN* resulted in increased Wnt reporter activity in HEK293STF cells in the absence of exogenous Wnt stimulation (Fig. 6b and Supplementary Fig. 6b). Importantly, this increased Wnt activity was not observed upon ectopic expression of CK1α binding-deficient *CRBN* mutants (Figs. 4d, 6b and Supplementary Fig. 6b). Furthermore, both Wnt3a and overexpressed *CRBN* resulted in the degradation of CK1α in these cells (Supplementary

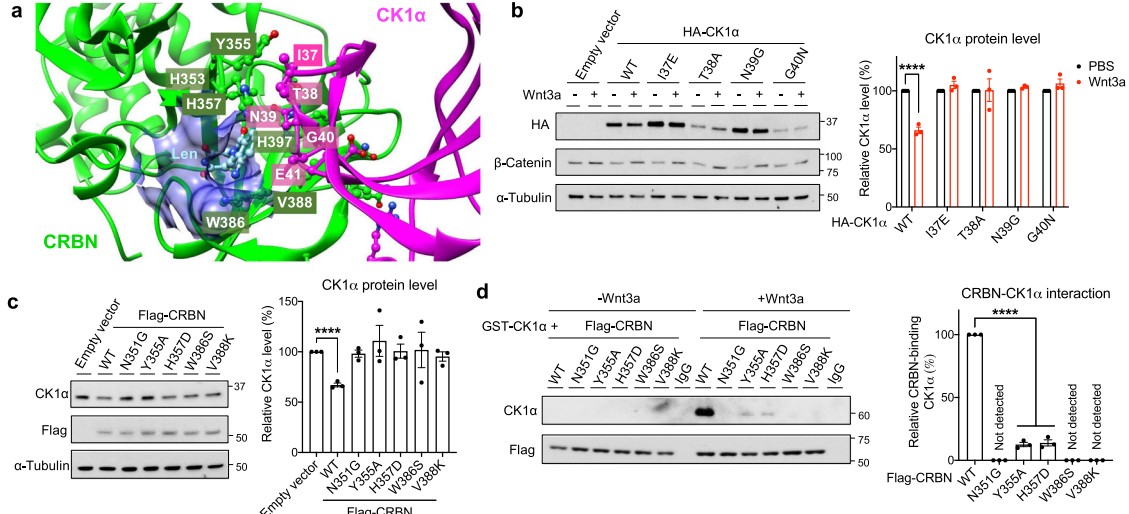

**Fig. 4 Wnt signaling induced CRBN-mediated degradation of CK1α requires its IMiD binding pocket. a** A structural model of CRBN-lenalidomide-CK1α complex (PDB: 5fqd). The interacting residues at the interface of the CRBN (green with white text) and CK1α (magenta with white text) are shown as a ball and stick model. The IMiD binding pocket is depicted in blue. **b** HEK cells were transfected with a plasmid encoding wild-type (WT) HA-tagged CK1α or an HA-tagged CK1α mutant, followed by PBS or Wnt3a treatment for 24 h. Extracts of these cells were evaluated by immunoblotting. A representative immunoblot (left panel) and a quantification of immunoblots (mean ± SEM, $n = 3$ independent experiments; right panel) are shown. **c** HEK cells were transfected with a plasmid encoding WT Flag-tagged CRBN or the indicated Flag-tagged CRBN mutants. Extracts of these cells were evaluated by immunoblotting. A representative immunoblot (left panel) and a quantification of immunoblots (mean ± SEM, $n = 3$ independent experiments; right panel) are shown. **d** HEK cells were transfected with a plasmid encoding WT Flag-tagged CRBN the indicated Flag-tagged CRBN mutants and subsequently treated with PBS or Wnt3a for 4 h. Flag-CRBN immunoprecipitated from extracts of these cells were incubated with the indicated amounts of recombinant GST-CK1α at 4 °C for 1 h. Flag-CRBN beads were re-isolated, washed, and CK1α bound was eluted using sample buffer. These immunoprecipitates were subjected to SDS-PAGE, followed by immunoblotting. IgG beads serve as a control for Flag-CRBN beads. A representative immunoblot (left panel) and a quantification of immunoblots (mean ± SEM, $n = 3$ independent experiments; right panel) are shown. Asterisks in **b**–**d** indicate statistical significance (two-tailed Student's $t$-test, $*p$ value < 0.05, $****p$ value < 0.0001).

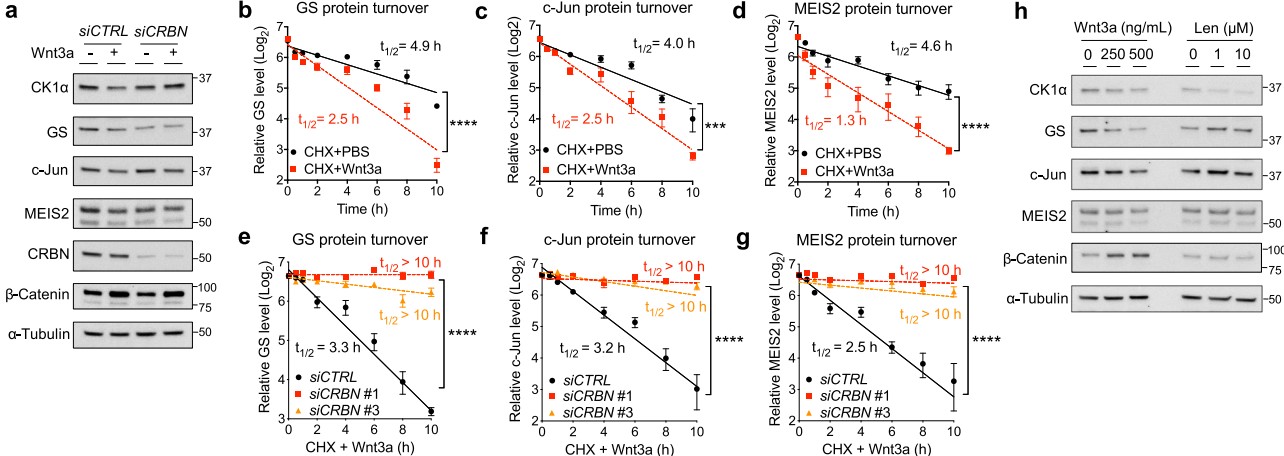

**Fig. 5 Wnt signaling induces CRBN-mediated degradation of a subset of endogenous substrates. a** HEK cells were transfected with the indicated smart-pool siRNA, followed by PBS or Wnt3a treatment for 24 h. Extracts of these cells were evaluated by immunoblotting. A representative immunoblot ($n = 3$ independent experiments) is shown. **b**–**d** HEK cells were co-treated with cycloheximide along with PBS or Wnt3a. Extracts of these cells were evaluated by immunoblotting. CK1α levels from immunoblots were quantitated, normalized to HSP90 levels, and plotted to determine the turnover of the indicated protein in response to Wnt3a (mean ± SEM, $n = 3$ independent experiments). **e**–**g** HEK cells were transfected with control (CTRL) or one of two distinct CRBN siRNA and co-treated with Wnt3a and cycloheximide. Extracts of these cells were evaluated by immunoblotting. CK1α levels from immunoblots were quantitated, normalized to HSP90 levels, and plotted to determine the turnover of the indicated protein in response to Wnt3a (mean ± SEM, $n = 3$ independent experiments). Asterisks in **b**–**g** indicate statistical significance (two-way ANOVA analysis, $***p$ value < 0.001, $****p$ value < 0.0001). **h** Extracts of HEK cells treated for 24 h with different doses of Wnt3a or lenalidomide were evaluated by immunoblotting. A representative immunoblot ($n = 3$ independent experiments) is shown.

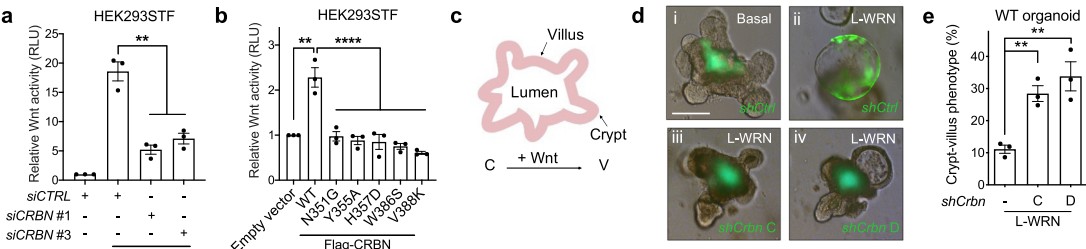

**Fig. 6 CRBN is a positive regulator of Wnt activity. a** The Wnt reporter gene expressing cell line, HEK293STF, was transfected with *control* (*CTRL*) siRNA or one of two distinct *CRBN* siRNA and treated with PBS or Wnt3a (100 ng/mL) for 48 h. Luciferase activity was subsequently determined and normalized to total protein concentration. A quantification of Wnt reporter activity (mean ± SEM, $n = 3$ independent experiments) is shown. **b** HEK293STF cells were transfected with a control plasmid or a plasmid encoding wild-type (WT) Flag-tagged CRBN, or the indicated Flag-tagged CRBN mutants for 48 h. Luciferase activity was determined and normalized to total protein concentration. A quantification of Wnt reporter activity (mean ± SEM, $n = 3$ independent experiments) is shown. **c** A schematic of a mouse intestinal organoid showing the presumptive crypt (C) and villus (V) regions. **d** Mouse intestinal organoids were infected with *control* (*Ctrl*) or one of two distinct *Crbn* shRNA (marked by GFP expression) and then cultured in (i) basal media or (ii–iv) 25% exogenous Wnt conditioned media (L-WRN) for 5 days. Representative images are shown ($n = 3$ independent experiments). Scale bar = 100 μm. **e**. Quantification of the percent of organoids that exhibit a more basal-like phenotype (mean ± SD, $n = 3$ technical replicates) in a representative experiment ($n = 3$ independent experiments) is shown. Asterisks in **a**, **b**, and **e** indicate statistical significance (two-tailed Student's *t*-test, **$p$ value < 0.01, ****$p$ value < 0.0001).

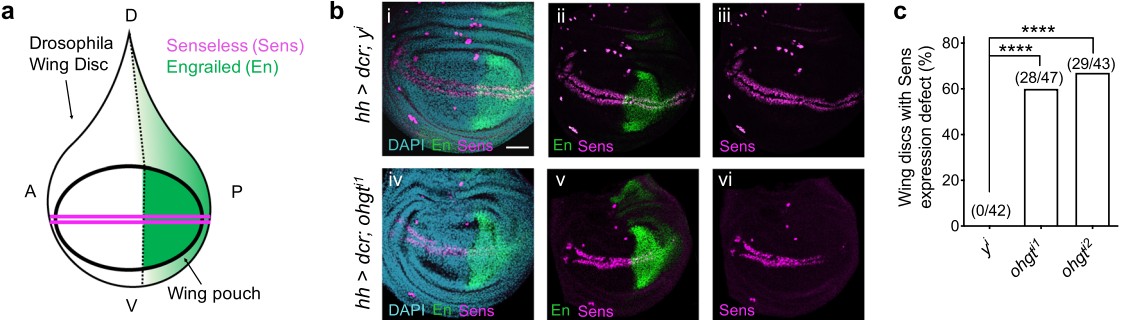

**Fig. 7 CRBN modulates Wnt function in Drosophila. a** A schematic showing a *Drosophila melanogaster* wing imaginal disc. A anterior; P posterior; D dorsal; V ventral. **b** Representative confocal images showing the level of the Wg/Wnt biomarker Senseless (Sens, magenta) after RNAi-mediated knockdown of (i–iii) yellow (y) control or (iv–vi) *ohgata* (*ohgt/crbn*), driven by a *hh-Gal4* driver in the posterior compartment of third instar wing imaginal disks. The region in which *hh-Gal4* drives expression is indicated by Engrailed (En) in green. DAPI staining (blue) is used to mark the wing disc. hh hedgehog, dcr dicer. Scale bar = 50 μm. **c** A quantification of wing disks with a defect in Sens levels is shown. Asterisks indicate statistical significance (Fisher's exact test, ****$p$ value < 0.0001).

Fig. 6a, b). These results are consistent with CRBN regulating Wnt activity in a manner dependent on CK1α.

Mouse intestinal organoids are grown in the presence of exogenous Wnt transform from a budded phenotype to a cystic morphology[41] (Fig. 6c, di, ii and Supplementary Fig. 6di). *Crbn* knockdown significantly attenuated this Wnt-stimulated cystic phenotype, consistent with an inhibition of Wnt activity (Fig. 6diii–iv, e and Supplementary Fig. 6dii-iii). We previously showed that CK1α levels are decreased in intestinal organoids supplemented with exogenous Wnt[15]. Herein, we find that knockdown of *Crbn* rescues this Wnt-driven decrease in CK1α levels (Supplementary Fig. 6c). Together, these results show that CRBN is required for Wnt signaling and that this function of CRBN occurs primarily via the regulation of CK1α.

**CRBN modulates Wnt function in vivo.** Wnt signaling is evolutionarily conserved across phyla, with many pivotal discoveries in the Wnt pathway originating from studies using *Drosophila melanogaster*[42]. Wnt/Wingless (Wg) signaling directs cell fate specification during the development of the wing imaginal disc, the precursor of the adult wing[43]. During this process, the Wg target gene *senseless* (*sens*) is transcriptionally activated on either

side of a row of Wg-expressing cells at the dorsoventral boundary[44] (Fig. 7a, bi–iii and Supplementary Fig. 7a, di–iii). We performed RNAi-mediated knockdown of the *CRBN* ortholog, *ohgata* (*ohgt*)[45], in the posterior compartment of the wing imaginal disc and observed significantly disrupted Sens expression across the posterior region of wing disks (Fig. 7biv–vi, c and Supplementary Fig. 7bi–iii, c, div–ix). Similar results were observed upon knockdown of the well-established regulator of Wg signaling *disheveled* (*dsh*)[46], albeit in a manner that was more penetrant (Supplementary Fig. 7biv–vi, c). Of note, the expression of Wg was not disrupted by *ohgt* knockdown (Supplementary Fig. 7a, di, ii, iv, v, vii, viii), indicating that Notch signaling, which is required for *wg* transcription at the dorsoventral margin[47], remained intact. These results provide evidence that Ohgt/Crbn promotes Wg/Wnt signal transduction in Drosophila.

Canonical Wnt signaling plays a pivotal role in midline patterning during zebrafish development, with aberrant Wnt activation resulting in eye loss[48] and Wnt/β-Catenin inhibition resulting in cyclopia[49–52] (Fig. 8a). We injected either *crbn* mRNA[53], *crbn* morpholino (MO)[35], or three distinct guide RNA (sgRNA) sequences targeting *crbn* along with *dCas9* mRNA (CRISPRi), into zebrafish embryos and examined the phenotype of these head structures. Increased expression of *crbn* mRNA

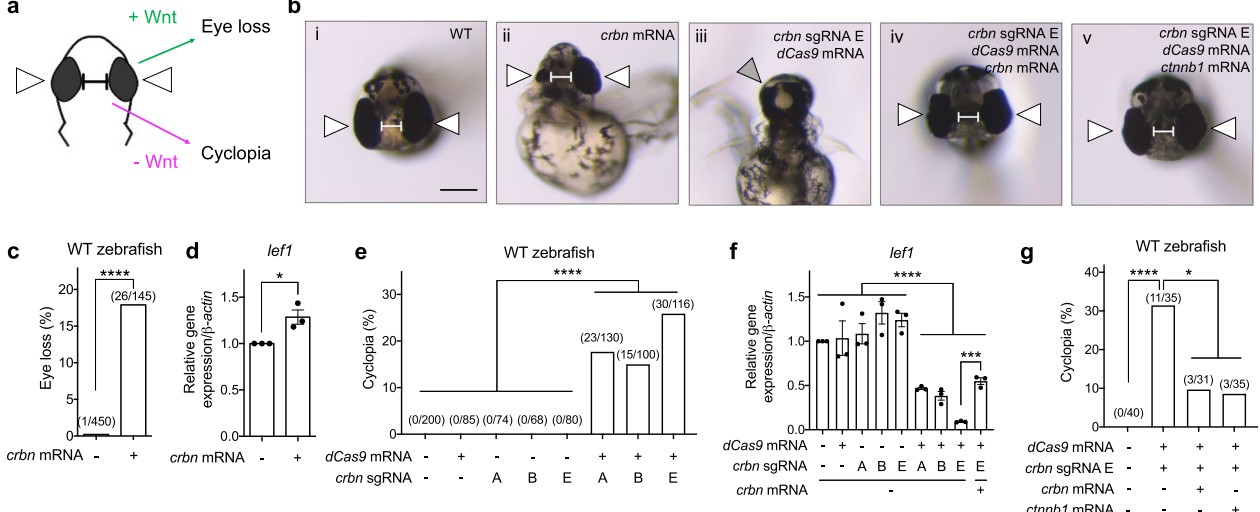

**Fig. 8 CRBN modulates Wnt function in zebrafish. a** A schematic of transverse views of a zebrafish head indicating phenotypes regulated by Wnt signaling. White arrows indicate eyes and the balbis indicates the distance between eyes. **b** The transverse views of a (i) WT zebrafish head, a WT zebrafish head injected with (ii) *crbn* mRNA, or (iii–v) one of three distinct *crbn* guide RNAs (sgRNA) along with *dCas9* mRNA, in the absence or presence of (iv) *crbn* mRNA or (v) *β-catenin* (*ctnnb1*) mRNA. White arrows indicate eyes, the gray arrow indicates merged eyes, and the balbis indicates the distance between eyes. Scale bar = 200 μm. **c**, **e** and **g** Quantifications of the indicated eye phenotypes are shown. Asterisks indicate statistical significance (Fisher's exact test, *$p$ value < 0.05, ****$p$ value < 0.0001). **d** and **f** RNA extracts from the indicated zebrafish (20 hpf) were used to determine the expression of three Wnt target genes using quantitative RT-PCR. Quantification of gene expression in indicated zebrafish embryos (mean ± SEM, $n = 3$ independent pools of zebrafish embryo) is shown. Asterisks indicate statistical significance (two-tailed Student's *t*-test, *$p$ value < 0.05, ***$p$ value < 0.001, ****$p$ value < 0.0001). lef1 lymphoid enhancer-binding factor 1.

resulted in eye loss (18%) (Fig. 8bii, c) relative to WT zebrafish (Fig. 8bi, c), whereas injection of *crbn* MO or CRISPRi constructs resulted in increased cyclopia (7% or 15–26%, respectively) (Supplementary Fig. 8ai, iv, b or Fig. 8biii, e, and Supplementary Fig. 8ei, ii) relative to their respective negative controls (Fig. 8bi and Supplementary Fig. 8aiii, b or Fig. 8bi, e, and Supplementary Fig. 8eiii–vi). Consistent with the cyclopic phenotype observed upon *crbn* knockdown (by MO or CRISPRi) being specific, co-injection with *crbn* mRNA significantly attenuated this phenotype (Fig. 8biv, g and Supplementary Fig. 8aii, b). Importantly, the *crbn* knockdown-induced cyclopic phenotype was rescued by co-expression of mRNA encoding β-catenin (Fig. 8bv, g). These results suggest that *crbn* positively modulates Wnt signaling in zebrafish. Consistently, corresponding changes in the expression of canonical Wnt/β-catenin target genes were observed upon modulation of *crbn* levels, correlating alterations of *crbn* levels with the regulation of canonical Wnt signaling in vivo (Fig. 8d, f and Supplementary Fig. 8c, d, f), further supporting the role of zebrafish *crbn* in canonical Wnt/β-catenin signaling. Moreover, we also noted that Ck1α protein levels in zebrafish were correspondingly regulated by *crbn* modulation (Supplementary Fig. 9). Together, these results show that CRBN regulates Wnt pathway activity in vivo, which leads to the Wnt-dependent modulation of CK1α abundance and Wnt-relevant phenotypes.

## Discussion

These studies show that CRBN is an evolutionarily conserved regulator of the Wnt signaling pathway, whose Wnt-dependent activation results in the ubiquitin-dependent, proteasomal degradation of a subset of substrates. Our results indicate that the regulation of CK1α protein levels is one of the major Wnt-driven functions of CRBN (Fig. 9)[21,40]. However, the degradation of other Wnt-driven CRBN substrates, such as those identified in our studies, may also represent important regulators of Wnt-driven biology. Our studies highlight an extracellular ligand-stimulated mechanism of regulation for CRBN, via modulation of Wnt activity.

Although we do not fully understand the mechanism by which Wnt regulates CRBN, we show that such regulation relies on the previously described IMiD binding pocket[21,40]. This finding underscores the significance of this structural motif for endogenous CRBN regulation, which we speculate may function as a binding pocket for an endogenous small-molecule CRBN regulator. Our findings may also have important clinical implications, as CRBN modulators might provide a potential form of pharmacological intervention in Wnt-driven diseases.

## Methods

**Reagents**. Recombinant Wnt3a (R&D Systems) was reconstituted with 0.1% BSA in PBS to 200 ng/μL (human Wnt3a) or to 40 ng/μL (mouse Wnt3a), snap-frozen and stored at −80 °C. Cycloheximide (Sigma), BA (Cell Signaling Technology), MG132 (Selleck Chemicals), lenalidomide (Sigma), the DUB inhibitor PR-619 (Selleck Chemicals), MLN4924 (Selleck Chemicals), and the ROCK inhibitor Y27632 (Stemcell Technologies) were dissolved in 100% DMSO at 3 to 50 mM and stored at −20 °C. Plasmids expressing human *Flag*-tagged *CRBN* (NP_001166953.1) and *HA*-tagged human *CK1α* (NP_001883.4) were synthesized in pcDNA3.1 (Addgene) and point mutations introduced by Genscript. Smart-pool siRNA targeting specific genes (*AXIN1*, *APC*, and *CRBN*) were purchased (Dharmacon). Individual siRNA targeting *CRBN* were purchased from Dharmacon (#1: J-021086-09-0005; #2: J-021086-10-0005; #3: J-021086-11-0005). Lentiviral expressing shRNA targeting human *CRBN* (#1: TL305228V #B; #2: TL205228V #D), mouse *Scramble control* (TR30021V), or mouse *Crbn* (#A-D: TL503372V #A, B, C, D) were purchased (Origene).

**Cell-based assays**. HEK293T (HEK), HEK293STF, NIH3T3, and L-WRN cells were purchased from American Type Culture Collection (ATCC) and maintained as recommended. Cells (50–70% confluent) were treated with recombinant Wnt3a ligands, the indicated drugs or siRNA in media containing 5% FBS. Wnt3a was added at 250 ng/mL to cells for 24 h unless otherwise indicated. About 50 nM of the indicated siRNA was transfected using Lipofectamine RNAiMAX (Thermo Fisher) following the manufacturer's protocol, for 48 h. Plasmids were transfected using Lipofectamine 2000 (Thermo Fisher) in media containing 10% FBS, for 48 h. Total RNA was extracted from cells using RNeasy plus kit (Qiagen), reverse transcribed using a high-capacity cDNA reverse transcription kit (Applied Biosystems), and used for quantitative PCR using specific TaqMan probes (Invitrogen) (Supplementary Table 1). RNA expression was then analyzed by Bio-Rad CFX manager

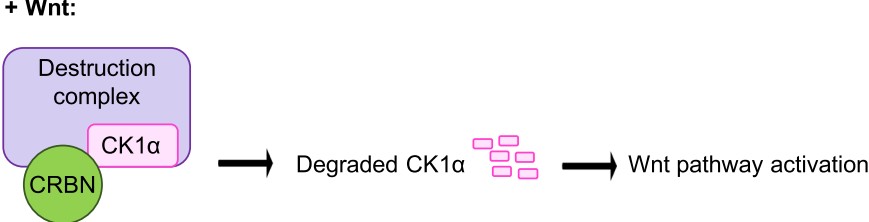

**Fig. 9 A model depicting Wnt-driven, CRBN-dependent regulation of CK1α.** Upon stimulation of Wnt signaling, CRBN is recruited to the β-Catenin destruction complex and targets CK1α for degradation, further promoting Wnt pathway activation.

3.1 software. For Wnt reporter assays, following treatment HEK293STF cells were lysed in 1X passive lysis buffer (Promega) at room temperature for 10 min, followed by a luciferase assay (Promega). Luminescence was recorded by a Veritas microplate reader. Cytoplasmic and nuclear cell fractions were separated using NE-PER™ Extraction Kit (Thermo Fisher) as per the manufacturer's protocol.

**Immunoblotting and immunoprecipitation**

*Immunoblotting.* Cells were lysed in an SDS sample buffer and boiled for 5 to 10 min, followed by SDS polyacrylamide gel electrophoresis. Proteins were then transferred to a nitrocellulose membrane and immunoblotted. The primary antibodies used were CK1α (1:1000 ab108296) and SIAH1 (1:250, ab2237) from Abcam, Axin1 (1:1000, 2087), APC (1:1000, 2504), GSK3β (1:1000, 9315), phosphorylated β-Catenin S45 (1:1000, 9564), phosphorylated β-Catenin S33, 37, T41 (1:1000, 9561), β-Catenin (1:1000, 9562), c-Myc (1:5000, 5605), GAPDH (1:5000, 8884), c-Jun (1:5000, 9165), PTEN (1:5000, 9552), and α-Tubulin (1:5000, 9099) from Cell Signaling Technology, α-Tubulin (1:5000, T6199) and GS (1:1000, MAB302) from Millipore, Ubiquitin (1:500, sc-8017) and HSP90 (1:5000, sc-13119) from Santa Cruz Biotechnology, P62 (1:1000, 610832) from BD Biosciences, CRBN (1:5000, NBP1-91810) and MEIS2 (1:250, H00004212-M01) from Novus Biologicals, and Flag (1:5000, F7425) from Sigma (Supplementary Table 2). The secondary antibodies used were HRP-conjugated donkey anti-mouse or anti-rabbit (715-035-150 or 711-035-152), or rabbit anti-goat IgG (305-025-045) from Jackson ImmunoResearch (1:10000). ImageJ was used to quantitate the signal of the indicated proteins in immunoblots.

*Immunoprecipitation.* Cells treated with PBS or Wnt3a for 4 h in the presence of 10 μM MG132 were lysed with Pierce IP lysis buffer (Thermo Fisher) containing a protease and phosphatase inhibitor cocktail (Thermo Fisher) and 10 μM MG132. For immunoprecipitation from HEK cells, the cell extracts were incubated with 1 μg of CRBN antibodies (Abcam ab68763) or PTEN antibodies (Santa Cruz Biotechnology sc-393186) overnight at 4 °C. Protein G agarose beads (Invitrogen) were then added for 1 h at 4 °C. For immunoprecipitation from NIH3T3 cells, CRBN antibodies (Cell Signaling Technology 71810) were first conjugated to agarose beads using an AminoLink Plus Immobilization Kit (Thermo Fisher) and these CRBN-conjugated beads were incubated in cell lysates overnight at 4 °C. Antibody-bound or -conjugated beads were then isolated by centrifugation and washed with lysis buffer for 10 min. For Flag immunoprecipitation, lysates were incubated with Flag antibody-conjugated beads (Sigma) overnight at 4 °C, and the beads were subsequently isolated by centrifugation and washed with lysis buffer for 10 min three times. Antibody-bound or -conjugated beads were resuspended in SDS sample buffer, boiled for 5 min, and used for the subsequent analysis.

**Ubiquitination assays.** HEK cells were treated with recombinant Wnt3a (100 ng/mL) and MG132 for the indicated amount of time. Cells were lysed with Pierce IP lysis buffer (Thermo Fisher), a protease and phosphatase inhibitor cocktail (Thermo Fisher), 50 μM PR-619, and 10 μM MG132. CK1α antibodies (Abcam ab206652) were conjugated to agarose beads using an AminoLink Plus Immobilization Kit (Thermo Fischer). The beads were incubated in cell lysates overnight at 4 °C, isolated by centrifugation, and washed with lysis buffer. For the tandem ubiquitin-binding entities (TUBE) assay, HEK cells were transfected with a plasmid expressing *HA-tagged CK1α* and then treated with recombinant Wnt3a (100 ng/ml) and MG132 for 6 h. Cells were then lysed as described above and incubated with TUBE-beads (Boston Biochem) for 2 h at 4 °C. The beads were subsequently isolated by centrifugation and washed with TBS containing 0.1% Tween-20 for 5 min three times. The antibody- or TUBE-conjugated beads were then boiled in an SDS sample buffer for 5 min and analyzed by immunoblotting.

For the in vitro ubiquitination reaction, HEK cells were transfected with a plasmid expressing *Flag-CRBN* and 48 h later treated with recombinant Wnt3a (100 ng/ml) and MG132 for 4 h. Cells were lysed and Flag-CRBN immunoprecipitated using Flag antibody-conjugated beads (Sigma) as described above. The beads were washed with lysis buffer three times and then once in E3 ligase buffer (Boston Biochem) at 4 °C. The following reagents were added to Flag-CRBN bound beads and 50 ng recombinant GST-CK1α (Invitrogen) in 1X E3 ligase buffer: 200 nM UBE1 (E1), 1 μM UBE2D1 (E2), 1 μM UBE2D3 (E2), 1 μM UBE2G1 (E2), 1 μg $K_0$ ubiquitin, 1 μM ubiquitin aldehyde, 10 μM MG132, and

50 μM PR-619, plus or minus 2.5 mM Mg-ATP (all from Boston Biochem). The reactions were mixed and then incubated at 30 °C for 3 h. The reactions were stopped by boiling with SDS sample buffer for 5 min, followed by immunoblotting.

**In vitro binding assay.** Flag-CRBN was immunoprecipitated from HEK cells, treated with or without recombinant Wnt3a, using Flag antibody-conjugated beads (Sigma). IgG-conjugated beads (Sigma) served as a negative control for Flag-CRBN isolation. The antibody-conjugated beads were then blocked with 10% BSA in 1 M Tris-HCl (pH 7.25) for 6 h at 4 °C as previously described[54]. The blocked beads were next washed with 1 M NaCl and subsequently incubated with recombinant GST-CK1α in binding buffer (25 mM HEPES pH 7.25, 100 mM NaCl, 0.01% Triton X-100, 1 mM DTT, and 5% glycerol) for 1 h at 4 °C. After one 5 min wash with binding buffer, the antibody-conjugated beads were resuspended in SDS sample buffer and boiled for 5 min, followed by silver staining (Thermo Scientific) or immunoblotting.

**Organoids.** Wild-type mouse intestinal organoids were isolated and maintained as previously described[15]. Prior to shRNA lentiviral infection, organoids were collected and digested by Gentle Cell Dissociation Reagent (Stemcell Technologies) at room temperature for 10 min, followed by washing with basal culture medium. About 10,000 dissociated organoid cells were then resuspended in 25% conditioned L-WRN (Wnt3a, Rspondin3, Noggin) media, produced from L-WRN cells according to ATCC's protocol, which contained lentivirus particles (MOI = 10), 8 μM polybrene and 10 μM Y27632. Organoid cells were inoculated with the virus using a centrifugation-based protocol: $600 \times g$ for 2 h at room temperature, followed by incubation for 1 h at 37 °C. The cells were again pelleted, washed with culture media, resuspended in Matrigel, plated, and overlaid with basal culture media or 25% L-WRN conditioned media for 5 days. Images were obtained using an Olympus IX51 inverted fluorescence microscope and Olympus CellSens software.

**Model organisms.** NHGRI-1 zebrafish embryos[55] (1 cell) were injected with 15 ng *crbn* morpholinos (MO) in 3 nL into the yolk or 1 ng *crbn* mRNA in 1 nL into the single cell. The sequence of *crbn* mRNA and MO are as previously described[35,53]. For CRISPR/dCas9 injections, embryos were injected with 250 pg *dCas9* mRNA or/and 500 pg *crbn* sgRNA. For rescue experiments, embryos were injected with 250 pg *dCas9* mRNA, 500 pg *crbn* sgRNA, and 250 pg mRNA encoding Crbn or β-Catenin. Sequences of *crbn* sgRNA (Alt-R® CRISPR-Cas9 guide RNA, Integrated DNA Technologies) are as following: #A: CACGGCUAUGGCUGCUGAGA GUUUUAGAGCUAGAAAUAGCAAGUUAAAAUAAGGCUAGUCCGUUAU CAACUUGAAAAAGUGGCACCGAGUCGGUGCUUUU; #B:GGAUUGUAAA CACACGGCUAGUUUUAGAGCUAGAAAUAGCAAGUUAAAAUAAGGCUA GUCCGUUAUCAACUUGAAAAAGUGGCACCGAGUCGGUGCUUUU; #E: CAGAGCGGAUUGUAAACACAGUUUUAGAGCUAGAAAUAGCAAGUU AAAAUAAGGCUAGUCCGUUAUCAACUUGAAAAAGUGGCACCGAGU CGGUGCUUUU. Embryos were raised, fixed, and phenotyped at 1 dpf or 2 dpf[56]. Bright field images of embryos were acquired using a Zeiss Stemi 2000-CS microscope with an Olympus DP72 camera. For qPCR analysis, cDNA samples were generated from 20 hpf embryos and cDNA expression analyzed using specific SYBR Green primers (Thermo Fisher) (Supplementary Table 1)[57,58], followed by analysis on a Quant Studio 5 real-time PCR system (Applied Biosystems)[56]. For immunoblotting, 15 hpf embryos were separated into 3–15 embryo pools, homogenized in RIPA buffer (Thermo Fisher) containing a protease and phosphatase inhibitor cocktail (Thermo Fisher) and 10 μM MG132 for 1 h at 4 °C. Lysates were cleared by centrifuging at $8000 \times g$ for 10 min at 4 °C and then boiled in SDS sample buffer for 5 min. After boiling, samples were sonicated for 30 s and used for immunoblotting.

The RNAi lines: *ohgt*[i1] (VDRC, #110809), *ohgt*[i2] (VDRC #40486)), *dsh*[i] (Bloomington Drosophila Stock Center, #31306), and *y*[i] (VDRC #106068) were expressed in Drosophila third instar male larval wing disks using the *hh-Gal4* driver[59] in conjunction with *UAS-dcr-2*[60] (Bloomington Drosophila Stock Center #25757). All crosses were reared at 25 °C. Drosophila wing disks were then used for immunohistochemistry[56]. Fluorescent images were obtained with a Nikon A1RSi confocal microscope. To quantitate Sens expression, the length of Sens expressed in

the posterior region was measured and normalized to the total length of the posterior compartment.

**Animal care**. Zebrafish studies were performed following the animal protocol of the University of Maryland's Institutional Animal Care and Use Committee.

**Statistics**. A minimum of three independent replicates were performed for each experiment. The error bars shown represent the standard error of the mean (SEM) of at least three independent experiments, except for organoid studies—which instead show the standard deviation (SD) of three technical replicates in one of three independent experiments. Statistical relevance was determined using a two-tailed Student's $t$-test, a two-way ANOVA (protein turnover comparisons), or a Fisher's exact test (Drosophila and zebrafish phenotype studies) using Prism Graphpad 9. Asterisks indicate statistical significance (*$p$ values ≤ 0.05; **$p$ values ≤ 0.01; ***$p$ values ≤ 0.001; ****$p$ values ≤ 0.0001).

**Reporting Summary**. Further information on research design is available in the Nature Research Reporting Summary linked to this article.

## Data availability
All data supporting the findings of this study are available in the paper and the Supplementary information file. Raw data and original gel images are included in the Source Data file. All other relevant data were available from the authors upon reasonable request. Source data are provided with this paper.

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

## Acknowledgements

We thank all members of the Robbins, Capobianco, Lee, and Ahmed laboratories for their insightful advice and discussion regarding this work. We also thank GUMC Computational Chemistry Shared Resources (CCSR) for structural insights in this work. This work was supported by the National Institutes of Health (NIH) Grants: R01CA219189 to D.J.R., R35GM122516 to E.L., R01GM121421, R01GM122222, and R35GM136233 to Y.A., R01CA244188 to D.J.R., E.L., and Y.A., and R01GM118557 to C.C.H. This work was also supported by the Dwoskin Cancer Fund and the National Cancer Institute of the NIH under Award Number P30CA240139 to Sylvester Comprehensive Cancer Center.

## Author contributions

C.S. and D.J.R. designed this project. E.L., Y.A., and N.G.A. provided insights with project design. C.S., A.N., L.R.N., A.A.A., M.S.-I., L.M.S., H.B., H.W., N.B., B.L., and S.D. obtained data and provided experimental support. C.S., A.N., L.R.N., A.A.A., L.M.S., H.B., N.B., S.D., and C.H.W. analyzed and interpreted the data. C.S. and D.J.R. wrote the manuscript. E.L., Y.A., N.G.A., B.L., H.B., N.B., C.C.H., and A.J.C. critically revised the manuscript. A.N., D.T.W., F.Y., and M.G.-C. proofread the manuscript.

## Competing interests

D.J.R., E.L., and A.J.C. are founders of StemSynergy Therapeutics Inc., a company commercializing small-molecule cell signaling inhibitors, including Wnt inhibitors. The remaining authors declare no competing interests.
