## [Peer Review File · Nature Communications]

REVIEWER COMMENTS

Reviewer #1 (Remarks to the Author):

The authors have two stories in one paper with a potential connection. They identify Cereblon (CRBN) as a regulator of Wnt/ β -catenin signaling in a cell line, organoids, drosophila and zebrafish. This part is strong and novel. However, it lacks robust mechanistic insight. In addition, they start the manuscript with the observation that CRBN can regulate the ubiquitylation and abundance of CK1 α , an important regulator of β -catenin S45 phosphorylation. There are several weak points in the connecting of the dots.

1. The CRBN knockdowns are done with a single siRNA without rescue. siRNA experiments need better controls.
2. The model predicts that effects of CRBN knockdown should be able to be rescued by over-expression of CK1 α , but this was not tested.
3. It's not clear to me that the effect of CRBN on CK1 α abundance is direct, versus indirect.
4. WNT3A seems to cause mono-, rather than poly-ubiquitylation of CK1 α (Figs 1G), and the evidence for direct action of CRBN on CK1 α is weak (Fig 2D).
5. How WNT3A signaling regulates association of CRBN with the destruction complex is unclear.
6. GSK3 inhibition, acting downstream of CK1 α , also causes a decrease in CK1 α protein. Is this regulated by CRBN?

Specific questions for the authors:

Fig 2D purports to show that adding Flag-CRBN in vitro "enhances" CK1 α ubiquitylation. This figure is not compelling data. Is this difference statistically significant? This seems like a key point to clarify, as the effect could readily be indirect.

The authors rely extensively on a single CRBN siRNA. siRNA has many off-target effects, and so needs to be better supported. Best practice is both use of multiple distinct siRNA sequences, and a rescue experiment with an siRNA 'immune' cDNA.

Fig 3A shows that CRBN co-precipitates with multiple components of the β -catenin destruction complex after WNT3A stimulation. This is really interesting, and of course raises the question of what other substrates does CRBN have in the complex, and how is this regulated. Some of this is beyond the scope of the paper, but raises the question of how direct the effect on CK1 is. This gets us to Fig 3B-D.

I'm not clear on what Fig 3B-D teach us about CRBN-CK1 α direct versus indirect. They tell us that when you knockdown the scaffold, the protein on the scaffold becomes less stable. The role of CRBN in this process is not addressed. I suggest these experiments could be expanded (e.g. knockdown AXIN1 or inhibit GSK3 +/- CRBN knockdown).

Most confusing to me, Fig 3D shows that inhibition of GSK3 stimulates CK1 α disappearance. Since GSK3 is downstream of CK1 α while WNT3A treatment is upstream, it's not clear to me what mechanism related to WNT3A and CRBN is implicated in CK1 α disappearance here. This needs to be clarified.

Fig 3E – the effect of CRBN knockdown is modest, and with p only 0.05, so this result is not compelling. First, please show all the data points and not bar graphs, as this is a key experiment. Second, as an experimental suggestion, the WNT3A stimulation might be maxed out and so reductions are harder to see. The reduction of signaling by CRBN knockdown might be better tested at sub-maximal (not 450-fold, but instead 20-fold) activation by WNT3A.

Line 171: "Wnt stimulation also decreased Ck1 α levels in organoids". This is an important implication

of the prior data and the results in Fig 3F. However, the data presented in extended data 3B, lane 2 compared with lane 1 does not strongly support that claim. I get that organoid westerns are difficult, but extended figure 3B is not strong. It is clear that shCrbnA-D makes CRBN decrease, and CK1 α increase.

Did the authors try to correlate the changes in CK1 α mediated by CRBN with changes in β -catenin S45 phosphorylation?

Minor points addressable in text.

Plunger plots. e.g. Figures 1D, 1E, 2E, 3E, 3Fvi, 4, should be replaced. Experiments with small samples sizes should use scatter-plots or similar methods that allow evaluation of the distribution of the data.

Fig 1 legend: "HEK cells were transfected with HA-tagged ubiquitin (HA-Ub)". I expect they mean that the cells were transfected with a plasmid driving expression of HA-tagged ubiquitin? Or did they transfect protein?

Figures 1F, 1G, 2D need molecular weight markers for proper interpretation. Fig 1F and G, is the predominant species mono-ubiquitylated CK1 α ? If it's mono-ubiquitylation, what are implications for proteasome? It would be interesting to compare the results of CK1 α with β -catenin in these assays, since β -catenin is polyubiquitylated inversely with CK1 α ubiquitylation.

Nomenclature point – human and mouse proteins are all caps, e.g. WNT3A. Wnt3a and Axin1 are not correct usage. Mouse proteins are also all caps, so mouse CK1 α protein (line 171), not Ck1 α

159. Line 160: "TOP/Flash Wnt reporter gene" should be TOPFlash (no /)

Extended Fig 3B: what do shCrbn A B C and D mean? Different shRNA sequences? Different experiments? Please clarify in text. I find only C and D in the Methods.

Any role for FAM83F in this process? C.f. <https://doi.org/10.1101/2020.05.25.114660>

Reviewer #2 (Remarks to the Author):

The manuscript by Shen et al. provides an interesting relationship between CRBN and Wnt-mediated signalings. After discovering CRBN as a target of thalidomide and its analogs, many reports have shown that CRBN works as a CRL4 E3 ligase and recognizes certain neosubstrates in the presence of various ligands such as thalidomide. However, the function of drug-unbound CRBN remains largely unclear. In this study, the authors suggest that Wnt3a induced the degradation of CK1 α by CRL4CRBN. The authors also found that CRBN interacted with the CK1 α -containing destruction complex and demonstrated that CRBN was required for Wnt signaling using the Organoid, Zebrafish, and Drosophila. The study may provide an important clue for the understanding of the original CRBN role in Wnt signaling. Therefore, the study is valuable, but there are many concerns, and the authors should address my comments listed below.

Major comments:

1. My biggest concern in this manuscript is that the authors have not clarified that the target of Wnt-dependent degradation by CRBN is indeed CK1 α and not other subunits of the destruction complex. In Figs. 3B and C, the authors showed that loss or inhibition of other subunits of the destruction complex

led to reduction of CK1a. There remains the possibility that CK1a is not a direct CRBN substrate. Decrease in CK1a might be resulted from the secondary effect of the breakdown of the real CRBN substrate. The authors should conduct the rescue experiments using a CRBN-binding defective mutant of CK1a and show it is sufficient to confer resistance to Wnt signaling by CRBN-mediated regulation.

2. The authors mentioned that Wnt signaling promotes degradation of a subset of protein substrates by stimulating CRBN activity (Figs. 1 and 2). However, it is unclear whether Wnt signaling increases CRBN activity or promotes susceptibility of the destruction complex to degradation by CRBN, or both.

3. The ubiquitination assays (Fig. 1F, G, and Fig. 2D) are lacking essential information and appropriate controls that are described below. (i) The positions of molecular weight markers are required. (ii) The blots of input are required. (iii) For in vivo ubiquitination assays, the blots of anti-ubiquitin (or anti-HA for Fig. 1F) are required. (iv) The appropriate negative controls are required. For Fig. 1F, I recommend HA-Ub minus with Wnt3a as a negative control. For Fig. 1F and G, it is better to perform these experiments using CRBN ^{-/-} cells. For Fig. 2D, I recommend ATP (-) as a control. (v) Fig. 2D also requires the properly exposed images of CK1a that confirm the same amount of CK1a protein subjected to each reactions. (vi) For Fig. 1F, it is difficult to compare the ubiquitination level of CK1a between 8 h and the others because that is combined image gathered from different locations or exposure times. The authors should provide the original images help to evaluate this result. (vii) I feel Fig. 1G and 2D are poor and need to be much cleaner to support author's claim.

4. The authors should validate the knockdown phenotypes using another siRNA or perform corresponding experiments using knockout cells. These are necessary to avoid off-target effects of siRNA.

5. In the Extended figure 2D, the authors used mouse NIH3T3 cells. Previous studies have shown that mouse CRBN did not bind any substrates such as Ikaros, Aiolos, and CK1a in the presence of lenalidomide because the critical valine was replaced with isoleucine (I391) in mouse CRBN. If Wnt signal induced the interaction between mouse CRBN and mouse CK1a, this is a very important finding for understanding the physiological role of CRBN. The authors should examine the interaction between mouse CRBN and mouse CK1a using mouse cells such as Figure 3A and also the effects of lenalidomide on the interaction.

6. In this manuscript, the role of the destruction complex for degrading CK1a proteins by Wnt treatment is unclear. How does the destruction complex contribute to CRBN-mediated Ck1a degradation?

7. In the final paragraph, the authors claimed that "these studies show that CRBN is a novel, evolutionarily conserved regulator of the Wnt signaling pathway, whose Wnt-dependent regulation results in the ubiquitin-dependent proteasomal degradation of a subset of substrates." However, the conservation is not proved in fly and zebrafish at the mechanistic level. In order to justify the present title of this manuscript, the authors should investigate whether the Crbn-dependent ubiquitination of CK1a accounts for the observed phenotypes in fly and zebrafish.

8. Exogenous Wnt stimulation attenuated intestinal organoid differentiation and knockdown of CRBN rescued the Wnt-induced pathological phenotype (Fig. 3F). The authors should consider gut organoid differentiation markers and examine whether CK1a expression affects the phenotype rescued by CRBN silencing.

Other comments:

1. Fig. 1B requires negative control, such as housekeeping genes.

2. In Fig. 2C, the data are preliminary, and the authors should conduct additional experiments. (1)

The immunoblot against CRBN is necessary. (2) The data do not rule out that CK1a was downregulated by the mechanism different from the authors' suggestion. To enhance the authors' hypothesis, the authors should check if the overexpression of CK1a-binding deficient CRBN mutants does not decrease CK1a.

3. The authors claimed that Wnt signaling induced CRBN-dependent degradation of other substrates (GS, c-jun, MEIS2), however authors only showed downregulation of the protein level by Wnt3a treatment. The authors should perform the RNA level analysis and half-life estimations.

4. What is "CRBN activity"? I have not found the definition of the activity in the manuscript. The authors should define it. To my knowledge, one of the functions of CRBN is to recognize the neosubstrate after binding to the ligand such as lenalidomide and dBET1 (Kronke et al. Nature 2015, Winter et al. Science 2015).

5. In fig 4A v-vii, some Sens signal is detectable in En+ area. The expression levels of Sens need to be quantitatively measured, and the quantification methods have to be described. It would be nice to have some control experiment where known Wnt regulator(s) is inactivated to evaluate the severity of the Crbn-knockdown phenotype. Scale bars are missing for all panels.

6. In fig 4B, according to the current guidelines for MO use in zebrafish (Stainier, PLoS Genet 2017), MOs need to be validated by using knock-out mutant. The validation is particularly important, when the authors describe a new phenotype that appears by MO injection. The authors should make sure that MO does not produce cyclopia when the MO is injected into crbn KO fish. It seems that the cyclopic phenotype was examined at 2 dpf. 2 dpf is too late as abnormalities in eye size and distance could arise as a secondary effect. These phenotypes would better be analyzed in early 1 dpf (for example, described in Pei et al., Developmental Biology 2009). Scale bars should be added to Bii-iv.

Reviewer #3 (Remarks to the Author):

In this study, Shen et al described an interesting role of CRBN in regulating WNT signaling. Authors showed that WNT recruits CRBN to the beta-catenin destruction complex and somehow activates the activity of CRBN. This leads to ubiquitination and degradation of CK1a and other known substrates of CRBN. Consistent with the negative role of CK1a in WNT signaling, CRBN-dependent degradation of CK1a promotes WNT/beta-catenin signaling. Loss of function studies in Drosophila and zebrafish support a role of CRBN in regulating WNT signaling in vivo. Overall, this is a nice study and it would certainly be interesting for people studying WNT signaling and CRBN. I have following suggestions for authors to improve the manuscript.

1. The finding that Wnt3a induces CK1a degradation is very interesting. However, Wnt3a recombinant protein is not 100% pure. To confirm the activity of Wnt3a recombinant protein is mediated by Wnt3a, authors should check whether FZD8-CRD can block the effect of Wnt3a recombinant protein on CK1a expression.

2. The extent of Wnt3a-induced CK1a degradation seems to vary in different experiments, at least based on quantifications provided in figures. Treatment of cells with 250ng/ml Wnt3a for 24 hrs decreased CK1a expression by 7 fold in Fig. 1A, but only 1 fold in Fig. 1E and Extended Fig. 1A. Authors should comment on this.

3. Fig. 2F. In CRBN-based glue degrader field, it is standard to use CRBN CRISPR KO cells to demonstrate the effect of glue degrader is mediated by CRBN. Authors should use HEK293 CRBN KO cells to demonstrate the effect of Wnt3a on CK1a and other CRBN substrates is dependent on endogenous CRBN.

4. Overexpression of CRBN in zebrafish induced eye loss (Fig. 4B). Does overexpression of CRBN increase WNT signaling in HEK293 cells?

5. Fig. 2C. Authors showed that overexpression of CRBN decreased the expression CK1a. Is this

mediated by increased ubiquitination and degradation of CK1a?

6. Petzold et al suggested that CRBN does not bind to CK1a in the absence of lenalidomide. Have authors tested the direct binding between CRBN and CK1a? What is the degron of CK1a that mediates WNT and CRBN-dependent degradation of CK1a? Is the same beta-hairpin loop of CK1a shown in the CK1a-lenalidomide-CRBN structure (Petzold et al) important for WNT and CRBN-dependent degradation of CK1a? Can authors mutate critical residues involved in CK1-CRBN binding based on the crystal structure?

7. The finding that Wnt3a increases the activity of CRBN is intriguing. Does Wnt3a affect post-translational modification of CRBN? Without solving the molecular mechanism, authors should at least provide some speculations in the discussion section.

Reviewer #1 (Remarks to the Author):

The authors have two stories in one paper with a potential connection. They identify Cereblon (CRBN) as a regulator of Wnt/ β -catenin signaling in a cell line, organoids, drosophila and zebrafish. This part is strong and novel. However, it lacks robust mechanistic insight. In addition, they start the manuscript with the observation that CRBN can regulate the ubiquitylation and abundance of CK1 α , an important regulator of β -catenin S45 phosphorylation. There are several weak points in the connecting of the dots.

We thank the reviewer for their insightful comments, which have led to a significantly improved manuscript. To better illustrate the connection between CRBN and CK1 α , we strengthened our data showing that CRBN ubiquitinates and degrades CK1 α in response to Wnt activation and validated that CRBN regulates Wnt signaling in a CK1 α -dependent manner across various model systems. Answers to their specific questions are shown below:

1. The CRBN knockdowns are done with a single siRNA without rescue. siRNA experiments need better controls.

In addition to previously used smart-pool siRNA, we have now used additional distinct individual siRNA or shRNAs targeting *CRBN*. These additional reagents were used to validate the effect of CRBN on CK1 α degradation and Wnt activity (see revised Figures 3B, 3D, 4E-G, 5A and Supplementary figures 2C-D, 2F-G, 4A, 4G, 5A).

2. The model predicts that effects of CRBN knockdown should be able to be rescued by over-expression of CK1 α , but this was not tested.

Our model predicts that CRBN degrades CK1 α . Therefore, its knockdown leads to increased levels of CK1 α and, as a result, decreased Wnt activity (see revised Figure 5A, 7A and Supplementary figures 5A, 7A), making the proposed experiment challenging. In contrast, our model also predicts that *CRBN* overexpression reduces CK1 α levels and hence decreases its ability to inhibit Wnt signaling (see revised Figures 5B, 7A and Supplementary figures 5B, 7A-B). Thus, we took advantage of this latter observation by using the established CK1 α agonist, pyrvinium, to rescue the effect of CRBN overexpression on Wnt activity. Such rescue experiments were successfully performed in both HEK293STF cells and in zebrafish (revised Figures 5B, 7Biii-iv, 7D).

3. It's not clear to me that the effect of CRBN on CK1 α abundance is direct, versus indirect.

To address whether the effect of CRBN on CK1 α abundance is direct, we performed a number of experiments:

a) We utilized an *in vitro* binding assay to show that CRBN isolated from Wnt-stimulated cells exhibits an increased ability to associate with purified recombinant CK1 α (revised Figures 2D-E).

- b) We utilized an *in vitro* ubiquitination assay to show that CRBN isolated from Wnt-stimulated cells exhibits an increased ability to ubiquitinate purified recombinant CK1 α (revised Figures 2D, 2F).
- c) We used a CK1 α mutant, which exhibits decreased binding to CRBN (doi: 10.1038/nature16979), to show that CK1 α is resistant to Wnt-mediated degradation when it cannot associate with CRBN (revised Figure 2C).
- d) Regarding arguments that CK1 α degradation might be a secondary effect due to the disruption of other destruction complex subunits, we now show that:
 - i. While Wnt3a induces a decrease in CK1 α levels within 6 hours, it has little effect on the steady-state levels of other components of the β -catenin destruction complex over the same period of time. This observation is not consistent with CRBN acting on CK1 α indirectly via disruption of the destruction complex (revised Figure 4A).
 - ii. Knockdown of *CRBN* rescues the Wnt-stimulated decrease in CK1 α protein levels without affecting the stability of other destruction complex components, inconsistent with CRBN acting on CK1 α indirectly via disruption of the destruction complex (revised Figure 4B).
 - iii. CK1 α degradation induced by disruption of the destruction complex (*APC* or *AXIN1* knockdown) remains CRBN-dependent (revised Figure 4D).

4. WNT3A seems to cause mono-, rather than poly-ubiquitylation of CK1 α (Figs 1G), and the evidence for direct action of CRBN on CK1 α is weak (Fig 2D).

We have optimized our CK1 α ubiquitination assays and validated that Wnt3a induces a polyubiquitination of CK1 α in cells (revised Figures 1F and Supplementary figure 1D). In addition, we now show that purified, recombinant CK1 α protein binds to and is ubiquitinated by Wnt-exposed CRBN *in vitro*, supporting the direct action of CRBN on CK1 α (revised Figures 2D-F). The answer to point #3 above also addresses the direct action of CRBN on CK1 α .

5. How WNT3A signaling regulates association of CRBN with the destruction complex is unclear.

We have shown that upon Wnt stimulation, CRBN is recruited to the destruction complex (revised Figures 4C). Further, we show that the Wnt stimulated degradation of CK1 α occurs prior to the dissociation of the destruction complex indicated by the level of scaffolding proteins Axin and APC (revised Figure 4A). However, when the integrity of the destruction complex is disrupted by the knockdown of *AXIN1* or *APC*, in the absence of Wnt3a stimulation, CK1 α is also degraded. This latter mechanism of CK1 α degradation is also CRBN-dependent (revised Figure 4D). To explain these various results, we now present a model in which the destruction complex protects CK1 α from CRBN induced degradation in the absence of Wnt. However, upon Wnt stimulation, the destruction complex serves as a scaffold to recruit CRBN, where it binds to CK1 α and ubiquitinates it (see revised Figure 8).

6. GSK3 inhibition, acting downstream of CK1 α , also causes a decrease in CK1 α protein. Is this regulated by CRBN?

Although GSK3 requires the upstream β -catenin phosphorylation mediated by CK1 α in order to phosphorylate β -catenin, GSK3 plays other roles in the destruction complex. For example, in the Wnt-off state, GSK3 regulates the interaction between Axin and APC to promote β -catenin destruction (doi: 10.7554/eLife.08022). In this case, there is no clear epistatic relationship between GSK3 and CK1 α . Therefore, we cannot ensure that the inhibition of GSK3 in our previous experiments occurs downstream of CK1 α . Thus, although we have new data showing that CK1 α degradation in response to GSK3 inhibition is attenuated by knockdown of CRBN, we no longer show any GSK3 inhibition data- in order to better focus our manuscript. If the reviewer preferred such data be included, we would be happy to add it back.

Specific questions for the authors:

1. Fig 2D purports to show that adding Flag-CRBN *in vitro* “enhances” CK1 α ubiquitylation. This figure is not compelling data...

We have further optimized our *in vitro* ubiquitination assay, which now better illustrates that Wnt-stimulated CRBN significantly increases the poly-ubiquitination of purified, recombinant CK1 α (revised Figure 2F).

2. The authors rely extensively on a single CRBN siRNA. siRNA has many off-target effects, and so needs to be better supported....

As suggested, we have now utilized multiple distinct siRNA and shRNA to target *CRBN*, confirming the effect of CRBN on CK1 α abundance and Wnt activity (revised Figures 3B, 3D, 4E-G, 5A and Supplementary figures 2C-D, 2F-G, 4A, 4G, 5A).

3. Fig 3A shows that CRBN co-precipitates with multiple components of the β -catenin destruction complex after WNT3A stimulation. This is really interesting, and of course, raises the question of what other substrates does CRBN have in the complex, and how is this regulated. Some of this is beyond the scope of the paper, but raises the question of how direct the effect on CK1 is. I’m not clear on what Fig 3B-D teach us about CRBN-CK1 α direct versus indirect. They tell us that when you knockdown the scaffold, the protein on the scaffold becomes less stable. The role of CRBN in this process is not addressed. I suggest these experiments could be expanded (e.g. knockdown AXIN1 or inhibit GSK3 +/- CRBN knockdown).

We now show that even though other destruction complex components are degraded by Wnt activation, only CK1 α degradation is CRBN-dependent (revised Figure 4B). These findings are consistent with CK1 α being the relevant CRBN substrate in the destruction complex. Also, as suggested, we now show that the CK1 α degradation resulting from the knockdown of *AXIN1* or *APC* is CRBN-dependent. These studies have helped clarify the role the destruction complex plays in CRBN mediated CK1 α degradation (see Figure 8).

4. Most confusing to me, Fig 3D shows that inhibition of GSK3 stimulates CK1 α disappearance.

Since GSK3 is downstream of CK1 α while WNT3A treatment is upstream, it's not clear to me what mechanism related to WNT3A and CRBN is implicated in CK1 α disappearance here. This needs to be clarified.

See the answer to point #6 above.

5. Fig 3E – the effect of CRBN knockdown is modest, and with p only 0.05, so this result is not compelling. First, please show all the data points and not bar graphs, as this is a key experiment...

In our revised Figure 5A, we now show that CRBN knockdown decreases Wnt reporter activity by approximately 75%, with a p value < 0.01 (**). As requested, we also now show all data points in the graph. In addition, we have clarified and expanded our definition of statistical significance in the methods.

6. Line 171: “Wnt stimulation also decreased Ck1 α levels in organoids”. This is an important implication of the prior data and the results in Fig 3F. However, the data presented in extended data 3B, lane 2 compared with lane 1 does not strongly support that claim. I get that organoid westerns are difficult, but extended figure 3B is not strong. It is clear that shCrbnA-D makes CRBN decrease, and CK1 α increase.

We have shown in our previous publication (doi: 10.1126/scisignal.aak9916) that Wnt signaling decreases the protein level of CK1 α in organoids, and now show that this Wnt-dependent CK1 α decrease can be rescued by knocking down *Crbn*. The relevant text has now been revised to clarify this point (Line 204-207).

7. Did the authors try to correlate the changes in CK1 α mediated by CRBN with changes in β -catenin S45 phosphorylation?

As now shown in revised Figure 4A, the change in CK1 α protein levels correlate with the change in β -Catenin S45 phosphorylation.

Minor points addressable in text.

1. Plunger plots. e.g. Figures 1D, 1E, 2E, 3E, 3Fvi, 4, should be replaced. Experiments with small samples sizes should use scatter-plots or similar methods that allow evaluation of the distribution of the data.

As suggested, we now show all data points in the relevant revised figures.

2. Fig 1 legend: “HEK cells were transfected with HA-tagged ubiquitin (HA-Ub)”. I expect they mean that the cells were transfected with a plasmid driving expression of HA-tagged ubiquitin?

We have corrected this mistake.

3. Figures 1F, 1G, 2D need molecular weight markers for proper interpretation. Fig 1F and G, is the predominant species mono-ubiquitylated CK1 α ? If it's mono-ubiquitylation, what are implications for proteasome? It would be interesting to compare the results of CK1 α with β -catenin in these assays, since β -catenin is polyubiquitylated inversely with CK1 α ubiquitylation.

We now show, in revised Figure 1F, 2F, and Supplementary figure 1D, that CK1 α is polyubiquitinated in response to Wnt stimulation. In addition, CK1 α is ubiquitinated in a manner that is inversely correlated with that of β -catenin (revised Supplementary figure 1D). In addition, we note that though a portion of CK1 α appears mono-ubiquitinated in our *in vitro* experiment (revised Figure 2F and Supplementary figure 1D), this was not the case in our experiments looking at ubiquitination of endogenous CK1 α (see Figure 1F).

4. Nomenclature point – human and mouse proteins are all caps, e.g. WNT3A. Wnt3a and Axin1 are not correct usage. Mouse proteins are also all caps, so mouse CK1 α protein (line 171), not Ck1 α

We have modified our nomenclature based on the guidelines provided by *Nature Communications*.

5. 159. Line 160: “TOP/Flash Wnt reporter gene” should be TOPFlash (no /).

As suggested, we have modified the text.

6. Extended Fig 3B: what do shCrbn A B C and D mean? Different shRNA sequences? Different experiments? Please clarify in text. I find only C and D in the Methods.

We thank the reviewer for pointing this issue out, which we have now clarified in the text, figure legend and methods.

7. Any role for FAM83F in this process?

While this is a very interesting question, in order to keep our manuscript focused, we have not addressed this question here.

Reviewer #2 (Remarks to the Author):

The manuscript by Shen et al. provides an interesting relationship between CRBN and Wnt-mediated signalings. After discovering CRBN as a target of thalidomide and its analogs, many reports have shown that CRBN works as a CRL4 E3 ligase and recognizes certain neosubstrates in the presence of various ligands such as thalidomide. However, the function of drug-unbound CRBN remains largely unclear. In this study, the authors suggest that Wnt3a induced the degradation of CK1a by CRL4CRBN. The authors also found that CRBN interacted with the CK1a-containing destruction complex and demonstrated that CRBN was required for Wnt signaling using the Organoid, Zebrafish, and *Drosophila*. The study may provide an important clue for the understanding of the original CRBN role in Wnt signaling. Therefore, the study is valuable, but there are many concerns, and the authors should address my comments listed below.

We thank the reviewer for their generous and insightful comments. To address their remaining questions, we have added a significant number of new and revised experiments, revised our text, and clarified various existing experiments. For responses to their specific comments, please see our responses below.

Major comments:

1. My biggest concern in this manuscript is that the authors have not clarified that the target of Wnt-dependent degradation by CRBN is indeed CK1a and not other subunits of the destruction complex. In Figs. 3B and C, the authors showed that loss or inhibition of other subunits of the destruction complex led to reduction of CK1a. There remains the possibility that CK1a is not a direct CRBN substrate. Decrease in CK1a might be resulted from the secondary effect of the breakdown of the real CRBN substrate. The authors should conduct the rescue experiments using a CRBN-binding defective mutant of CK1a and show it is sufficient to confer resistance to Wnt signaling by CRBN-mediated regulation.

To address this important question, we now show that:

- a) While Wnt3a induces a decrease in CK1 α levels within a 6-hour period of time, it has little effect on the steady-state levels of other components of the destruction complex over the same period of time. This observation is not consistent with CRBN acting on CK1 α indirectly via degradation of other components of the destruction complex (revised Figure 4A).
- b) Knockdown of CRBN prevents the Wnt-driven decrease in CK1 α protein levels but does not affect the stability of other destruction complex components. This finding is also inconsistent with CRBN acting on CK1 α indirectly via degradation of other components of the destruction complex (revised Figure 4B).
- c) CK1 α degradation induced by disruption of the destruction complex (*APC* or *AXIN1* knockdown) remains CRBN-dependent (revised Figure 4B).
- d) We also used a CK1 α mutant, which exhibits decreased binding to CRBN (doi: 10.1038/nature16979) to show that CK1 α is resistant to Wnt-mediated degradation when it cannot associate with CRBN (revised Figure 2C).

2. The authors mentioned that Wnt signaling promotes degradation of a subset of protein substrates by stimulating CRBN activity (Figs. 1 and 2). However, it is unclear whether Wnt signaling increases CRBN activity or promotes susceptibility of the destruction complex to degradation by CRBN, or both.

We have shown that upon Wnt stimulation, CRBN is recruited to the destruction complex (revised Figure 4C). Further, we show that the Wnt-stimulated degradation of CK1 α occurs prior to the dissociation of the destruction complex indicated by the level of scaffolding proteins Axin and APC (revised Figure 4A). However, when the integrity of the destruction complex is disrupted by the knockdown of *AXIN1* or *APC*, CK1 α is degraded in a Wnt3a-independent manner. This latter mechanism of CK1 α degradation is however still CRBN-dependent (revised Figure 4D). To explain these various results, we now present a model in which the destruction complex protects CK1 α from being destabilized by CRBN in the absence of Wnt. However, upon Wnt stimulation, the destruction complex serves as a scaffold to recruit CRBN, where it binds to CK1 α and ubiquitinates it (see revised Figure 8).

3. The ubiquitination assays (Fig. 1F, G, and Fig. 2D) are lacking essential information and appropriate controls that are described below. (i) The positions of molecular weight markers are required. (ii) The blots of input are required. (iii) For in vivo ubiquitination assays, the blots of anti-ubiquitin (or anti-HA for Fig. 1F) are required. (iv) The appropriate negative controls are required. For Fig. 1F, I recommend HA-Ub minus with Wnt3a as a negative control. For Fig. 1F and G, it is better to perform these experiments using CRBN $-/-$ cells. For Fig. 2D, I recommend ATP (-) as a control. (v) Fig. 2D also requires the properly exposed images of CK1a that confirm the same amount of CK1a protein subjected to each reactions. (vi) For Fig. 1F, it is difficult to compare the ubiquitination level of CK1a between 8 h and the others because that is combined image gathered from different locations or exposure times. The authors should provide the original images help to evaluate this result. (vii) I feel Fig. 1G and 2D are poor and need to be much cleaner to support author's claim.

We thank the reviewer for their comments and suggestions. We have now modified our ubiquitination assays and data presentation as suggested (revised Figures 1F, 2F and Supplementary Figure 1D).

4. The authors should validate the knockdown phenotypes using another siRNA or perform corresponding experiments using knockout cells. These are necessary to avoid off-target effects of siRNA.

In addition to previously used smart-pool *CRBN* siRNA, we now show results using additional distinct siRNA and shRNA targeting *CRBN*, which further validate the effect of CRBN on CK1 α degradation and Wnt activity (revised Figures 3B, 3D, 4E-G, 5A and Supplementary figures 2C-D, 2F-G, 4A, 4G, 5A).

5. In the Extended figure 2D, the authors used mouse NIH3T3 cells. Previous studies have shown that mouse CRBN did not bind any substrates such as Ikaros, Aiolos, and CK1a in the presence

of lenalidomide because the critical valine was replaced with isoleucine (I391) in mouse CRBN. If Wnt signal induced the interaction between mouse CRBN and mouse CK1a, this is a very important finding for understanding the physiological role of CRBN. The authors should examine the interaction between mouse CRBN and mouse CK1a using mouse cells such as Figure 3A and also the effects of lenalidomide on the interaction.

We now show that Wnt3a induces an interaction between CRBN and CK1 α in mouse fibroblasts, while lenalidomide does not (revised Supplementary figure 3B).

6. In this manuscript, the role of the destruction complex for degrading CK1a proteins by Wnt treatment is unclear. How does the destruction complex contribute to CRBN-mediated Ck1a degradation?

We have shown that upon Wnt stimulation, CRBN is recruited to the destruction complex (revised Figure 4C). Further, we show that the Wnt stimulated degradation of CK1 α occurs prior to the dissociation of the destruction complex indicated by the level of scaffolding proteins Axin and APC (revised Figure 4A). However, when the integrity of the destruction complex is disrupted by the knockdown of *AXIN1* or *APC*, CK1 α is degraded in a Wnt3a-independent manner. This latter mechanism of CK1 α degradation is, however, still CRBN-dependent (revised Figure 4D). To explain these various results, we now present a model in which the destruction complex protects CK1 α from being destabilized by CRBN in the absence of Wnt. However, upon Wnt stimulation, the destruction complex serves as a scaffold to recruit CRBN, where it binds to CK1 α and ubiquitinates it (see revised Figure 8).

7. In the final paragraph, the authors claimed that "these studies show that CRBN is a novel, evolutionarily conserved regulator of the Wnt signaling pathway, whose Wnt-dependent regulation results in the ubiquitin-dependent proteasomal degradation of a subset of substrates." However, the conservation is not proved in fly and zebrafish at the mechanistic level...

As suggested, we conducted additional experiments to support the stated conservation of function. We injected zebrafish embryos with *crbn* mRNA and showed corresponding changes in Ck1 α protein levels (revised Figure 7E and Supplementary figure 7C). In addition, we now show that the established Ck1 α agonist, pyrvinium, can rescue the eye loss phenotype induced by *crbn* mRNA (revised Figure 7Biii-iv, 7D, and Supplementary figure 7Aii-iv). Together, these experiments are consistent with *Crbn* primarily regulating Wnt activity via destabilization of Ck1 α in zebrafish.

Other comments:

1. Fig. 1B requires negative control, such as housekeeping genes.

In the original figure, gene expression of the indicated genes were already normalized to that of the housekeeping gene *GAPDH*, indicated in the y axis legend. We now have added *TBP*

expression as an additional housekeeping control (revised Figure 1B) and revised the legend to clarify this point.

2. In Fig. 2C, the data are preliminary, and the authors should conduct additional experiments. (1) The immunoblot against CRBN is necessary. (2) The data do not rule out that CK1a was downregulated by the mechanism different from the authors' suggestion. To enhance the authors' hypothesis, the authors should check if the overexpression of CK1a-binding deficient CRBN mutants does not decrease CK1a.

As another reviewer commented that this figure lacked some controls, we no longer show this figure. Instead, as suggested, we have used a CK1 α mutant that exhibits decreased binding to CRBN (doi: 10.1038/nature16979) and now show that this mutant is not degraded by Wnt3a (revised Figure 2C).

3. The authors claimed that Wnt signaling induced CRBN-dependent degradation of other substrates (GS, c-jun, MEIS2), however authors only showed downregulation of the protein level by Wnt3a treatment. The authors should perform the RNA level analysis and half-life estimations.

We have performed the analyses as suggested and now show that Wnt signaling significantly decreases the half-life of these other CRBN substrates, while the expression of their respective genes remained unchanged (revised Figures 3B-D and Supplementary figures 4B-F). In addition, we show that the alterations in protein half-life of these other substrates are CRBN-dependent (revised Figures 3E-G and Supplementary figures 4G).

4. What is "CRBN activity"? I have not found the definition of the activity in the manuscript. The authors should define it. To my knowledge, one of the functions of CRBN is to recognize the neosubstrate after binding to the ligand such as lenalidomide and dBET1 (Kronke et al. Nature 2015, Winter et al. Science 2015).

As suggested, we have modified the text to clarify this definition (Line 1, 80-81, 118-119, 125, 146-148).

5. In fig 4A v-vii, some Sens signal is detectable in En+ area The expression levels of Sens need to be quantitatively measured, and the quantification methods have to be described. It would be nice to have some control experiment where known Wnt regulator(s) is inactivated to evaluate the severity of the Crbn-knockdown phenotype. Scale bars are missing for all panels.

As requested, we added quantitation of Sens reduction following RNAi-mediated knockdown of *ohgata*, or the negative control gene (*yellow*), in the posterior compartment of the wing disc (revised Supplementary figure 6D). To quantitate, we measured the length of the posterior region in which Sens was reduced as a fraction of the total length of the posterior compartment. The quantitation method is described in the methods and the associated figure legends. In addition, as requested, we provide new RNAi-mediated knockdown data of *dishevelled*, an essential Wingless pathway component (revised Supplementary figure 6B). The knockdown of *dishevelled*

also decreased Sens in the posterior compartment, with more penetrant phenotypes compared to *ohgt* knockdown (revised Supplementary figure 6D). Scale bars have been added.

6. In fig 4B, according to the current guidelines for MO use in zebrafish (Stainier, PLoS Genet 2017), MOs need to be validated by using knock-out mutant. The validation is particularly important, when the authors describe a new phenotype that appears by MO injection. The authors should make sure that MO does not produce cyclopia when the MO is injected into *crbn* KO fish. It seems that the cyclopic phenotype was examined at 2 dpf. 2 dpf is too late as abnormalities in eye size and distance could arise as a secondary effect. These phenotypes would better be analyzed in early 1 dpf (for example, described in Pei et al., Developmental Biology 2009). Scale bars should be added to Bii-iv.

To validate our *crbn* MO-induced cyclopia phenotype we rescued the phenotype via the co-expression of *crbn* mRNA (revised Figure 7C and Supplementary figure 7Ai). Consistent with the *crbn* MO acting in a specific manner, zebrafish co-injected with *crbn* mRNA and MO do not exhibit any eye phenotype (revised Figure 7C and Supplementary figure 7Ai). In addition, we have now added 1 dpf images for WT and *crbn* MO phenotypes (revised Supplementary figure 7B). Scale bars have also been added to figures as suggested.

Reviewer #3 (Remarks to the Author):

In this study, Shen et al described an interesting role of CRBN in regulating WNT signaling. Authors showed that WNT recruits CRBN to the beta-catenin destruction complex and somehow activates the activity of CRBN. This leads to ubiquitination and degradation of CK1a and other known substrates of CRBN. Consistent with the negative role of CK1a in WNT signaling, CRBN-dependent degradation of CK1a promotes WNT/beta-catenin signaling. Loss of function studies in *Drosophila* and zebrafish support a role of CRBN in regulating WNT signaling in vivo. Overall, this is a nice study and it would certainly be interesting for people studying WNT signaling and CRBN. I have following suggestions for authors to improve the manuscript.

We thank the reviewer for their supportive and insightful comments. As suggested, we have improved our analyses and modified our text accordingly. For specific comments, please see our responses below.

1. The finding that Wnt3a induces CK1a degradation is very interesting. However, Wnt3a recombinant protein is not 100% pure. To confirm the activity of Wnt3a recombinant protein is mediated by Wnt3a, authors should check whether FZD8-CRD can block the effect of Wnt3a recombinant protein on CK1a expression.

As suggested, we co-treated cells with recombinant Wnt3a and FZD8-CRD and now show that FZD8-CRD successfully inhibits the CK1 α degradation induced by Wnt3a, supporting the specificity of Wnt3a treatment (revised Supplementary figure 1B).

2. The extent of Wnt3a-induced CK1a degradation seems to vary in different experiments, at least based on quantifications provided in figures. Treatment of cells with 250ng/ml Wnt3a for 24 hrs decreased CK1a expression by 7 fold in Fig. 1A, but only 1 fold in Fig. 1E and Extended Fig. 1A. Authors should comment on this.

The reviewer is correct in noting that although we always see decreases in CK1 α in response to Wnt3a, there is some variation in the extent of this decrease. For the most part, the reasons behind these variations are not clear to us but likely depend on the batch and activity of recombinant Wnt3a and levels of the confluence of the cells used. We have modified the text to note this variance (Line 275).

3. Fig. 2F. In CRBN-based glue degrader field, it is standard to use CRBN CRISPR KO cells to demonstrate the effect of glue degrader is mediated by CRBN. Authors should use HEK293 CRBN KO cells to demonstrate the effect of Wnt3a on CK1a and other CRBN substrates is dependent on endogenous CRBN.

We requested HEK *CRBN* KO cell lines from two different groups but have yet to receive them. Alternatively, we generated and used our own HEK *shCRBN* knockdown cell lines. These cell

lines allowed us to further demonstrate that CK1 α is not degraded by Wnt3a in the absence of CRBN (revised Supplementary figure 2C).

4. Overexpression of CRBN in zebrafish induced eye loss (Fig. 4B). Does overexpression of CRBN increase WNT signaling in HEK293 cells?

We now show that overexpression of CRBN increases Wnt reporter activity in HEK293STF cells (revised Figure 5B) and does so in a manner that can be rescued using the established CK1 α agonist, pyruvium. Together, these results further support that CRBN regulates Wnt activity in a CK1 α -dependent manner.

5. Fig. 2C. Authors showed that overexpression of CRBN decreased the expression CK1a. Is this mediated by increased ubiquitination and degradation of CK1a?

As another reviewer commented that this figure lacked some controls, we no longer show this figure. Instead, we have focused on validating our finding that CRBN ubiquitinates and degrades CK1 α by refining our existing assays and adding additional control experiments (revised Figures 1F, 2C, 2F, 7E and Supplementary figures 1D, 2C-D, 2F-G, 7C).

6. Petzold et al suggested that CRBN does not bind to CK1a in the absence of lenalidomide. Have authors tested the direct binding between CRBN and CK1a? What is the degron of CK1a that mediates WNT and CRBN-dependent degradation of CK1a? Is the same beta-hairpin loop of CK1a shown in the CK1a-lenalidomide-CRBN structure (Petzold et al) important for WNT and CRBN-dependent degradation of CK1a? Can authors mutate critical residues involved in CK1-CRBN binding based on the crystal structure?

a) We have performed an *in vitro* binding assay using Flag-CRBN purified from cells treated with PBS or Wnt3a and recombinant CK1 α protein. We now show that CRBN associates with CK1 α *in vitro* but that this interaction requires Wnt activation (revised Figure 2D-E). However, it remains possible that a bridging molecule is co-purified with CRBN and acts to facilitate CRBN and CK1 α binding. This result, along with this caveat, are now shown and discussed in the revised text (Line 139-142).

b) We also mutated one of the critical residues in CK1 α involved in lenalidomide-induced CRBN-CK1 α binding, based on the Petzold crystal structure, Gly40. We note that this CK1 α mutant is also resistant to Wnt-induced degradation. These results are consistent with the Gly40 beta-hairpin loop also being important for Wnt-driven degradation of CK1 α (revised Figure 2C).

7. The finding that Wnt3a increases the activity of CRBN is intriguing. Does Wnt3a affect post-translational modification of CRBN? Without solving the molecular mechanism, authors should at least provide some speculations in the discussion section.

We have not noticed a significant change in gene expression or protein levels of CRBN in response to Wnt activation. However, we did observe increased binding of CRBN to CK1 α and its subsequent ubiquitination in response to Wnt induction. Based on such results, we speculate that Wnt likely regulates CRBN via some post-translational mechanism. As CRBN has been reported to autoubiquitinate in the presence of some immunomodulatory drugs (10.1038/nature13527), we have added text in the discussion speculating that Wnt3a might regulate CRBN autoubiquitination in order to activate it (Line 251-253).

REVIEWER COMMENTS

Reviewer #1 (Remarks to the Author):

The authors have nicely addressed the issues raised, and increased the robustness of their analysis.

One last comment - the word 'novel' appears four times in the text. 'Novel' never ages well. I suggest rewording.

Reviewer #2 (Remarks to the Author):

The revised manuscript has been substantially improved, except for the second half, the part with drosophila and zebrafish. I have following suggestions and comments for authors to improve the manuscript.

1. I have a significant concern about Figure 7 and Supplementary figure 7 related to cyclopia. The authors have been used cyclopia as a readout of reduction of Wnt activity by citing ref #36 (Musso et al. Development 2014). However, in this reference, cyclopia is caused by convergence-extension defects due to Wnt11-deficiency, mediated by non-canonical Wnt-pathway and independent of β -catenin function (Heisenberg et al. Nature 2000), which makes me worry that effects of Crbn on β -catenin/destruction complex through Wnt3a cannot be evaluated by "cyclopia". And I think it possible that the smaller head size caused by crbn MO begets the loss of eyes or small eyes, regardless of Ck1a (Figure 7Bii).
2. The authors should consider the dose effects of pyrvinium, and present data on how the interaction between CRBN and Ck1a is altered by pyrvinium.
3. Blots in Supplementary figure 7C is not convincing in that it is not clear whether crbn MO stabilized Ck1a and the stabilization was cancelled by crbn mRNA. The authors should increase the number of samples and clearly show the statistical significance.
4. I would strongly encourage the authors to create crbn knock-out fish and validate the MO phenotypes. Recent studies using zebrafish explicitly demonstrated that describing novel phenotypes only with unvalidated MOs should be avoided whenever possible (Tessadori et al. Nature 2020, Jiang et al. Development 2020).

Reviewer #3 (Remarks to the Author):

Authors have largely addressed my points. However, CK1 and GSK3 play important roles in many biological processes. I would guess only a small pool of CK1 and GSK3 is associated with the Axin complex. It is possible that this pool of CK1 and GSK3 is regulated by WNT signaling. However, it does not seem to make sense that the total pool of CK1 is regulated by WNT. The molecular mechanism of WNT/CRBN mediated degradation of CK1a is also not clear.

REVIEWER COMMENTS

Reviewer #1 (Remarks to the Author):

The authors have nicely addressed the issues raised, and increased the robustness of their analysis.

One last comment - the word 'novel' appears four times in the text. 'Novel' never ages well. I suggest rewording.

We thank the reviewer for the comments. We have reworded the text as suggested.

Reviewer #2 (Remarks to the Author):

The revised manuscript has been substantially improved, except for the second half, the part with drosophila and zebrafish. I have following suggestions and comments for authors to improve the manuscript.

1.a. The authors have been used cyclopia as a readout of reduction of Wnt activity by citing ref #36 (Musso et al. Development 2014). However, in this reference, cyclopia is caused by convergence-extension defects due to Wnt11-deficiency, mediated by non-canonical Wnt-pathway and independent of β -catenin function (Heisenberg et al. Nature 2000).... “

We apologize for this incorrect reference. We now provide four distinct references showing that that inhibition of canonical Wnt signaling can lead to cyclopia (doi: 10.1006/dbio.1999.9537; 10.1242/dev.00402; 10.1016/s0925-4773(99)00319-6; 10.1242/dev.02295).

We also now provide data showing that *crbn* knockdown-induced cyclopia can be rescued by β -catenin/*ctnnb1* mRNA (Fig. 8Bv, 8G). Further, we show that the levels of canonical Wnt target genes are significantly reduced in *crbn* knockdown zebrafish (Fig. 8D, 8F and Supplementary fig. 8C-D, 8F). Together, these data support a model whereby the cyclopic phenotype observed upon *crbn* knockdown results from inhibition of canonical Wnt/ β -catenin activity.

1.b. I think it possible that the smaller head size caused by *crbn* MO begets the loss of eyes or small eyes, regardless of Ck1a.

As loss of eyes or small eyes are a phenotype associated with Wnt gain of function, which we see when *crbn* mRNA is expressed in zebrafish, we assume the Reviewer was actually referring to our *crbn* mRNA experiments? In such experiment, we **did not** observe any change in head size even though the zebrafish exhibited a corresponding loss/reduction of eyes (Fig 8Bii). However, as microcephaly is commonly associated with cyclopia (doi: 10.1136/bcr-2017-220159), we **did** observe a smaller head size in cyclopic zebrafishes injected with *crbn* targeted MO or CRISPRi. We were able to rescue the cyclopia in these zebrafishes using *crbn* mRNA or β -catenin mRNA, consistent with this phenotype resulting from decreased *crbn* expression attenuating canonical Wnt activity.

2. The authors should consider the dose effects of pyrvinium...”

We substantially revised our zebrafish data, validating our previous MO data with CRISPRi (three distinct guide RNAs), rescuing the resulting knockdown phenotype with β -catenin expression, and showing significant changes in Ck1 α protein levels. This supports our main conclusion that CRBN plays an essential role in canonical Wnt signaling *in vivo*. Thus, while still very interesting to us, we have removed our pyrvinium rescue data as it lies outside of this focus.

3. Blots in Supplementary figure 7C is not convincing in that it is not clear whether *crbn* MO stabilized Ck1a and the stabilization was cancelled by *crbn* mRNA. The authors should increase the number of samples and clearly show the statistical significance.

As suggested, we have performed additional immunoblotting analysis using the lysates from zebrafish embryos injected with *crbn* CRISPRi and now show that Ck1 α levels are significantly increased in such embryos (Supplementary fig. 9).

4. I would strongly encourage the authors to create *crbn* knock-out fish and validate the MO

phenotypes. Recent studies using zebrafish explicitly demonstrated that describing novel phenotypes only with unvalidated MOs should be avoided whenever possible.

Based on recently published guidelines on the use of MO in zebrafish (doi: 10.1371/journal.pgen.1007000), we have now validated our MO experiments using CRISPRi-targeting *crbn* with three distinct guide RNAs. These results are consistent with reduction of *crbn* expression resulting in cyclopia (Fig. 8Biii, 8E and Supplementary fig. 8A-B, 8E). We further validate this phenotype using rescue experiments with *crbn* mRNA and β -*catenin* mRNA (Fig. 8Biv-v, 8G). Combined, our results support a model in which reduction in Crbn decreases canonical Wnt-driven activity, resulting in cyclopia.

Reviewer #3 (Remarks to the Author):

Authors have largely addressed my points. However, CK1 and GSK3 play important roles in many biological processes. I would guess only a small pool of CK1 and GSK3 is associated with the Axin complex. It is possible that this pool of CK1 and GSK3 is regulated by WNT signaling. However, it does not seem to make sense that the total pool of CK1 is regulated by WNT. The molecular mechanism of WNT/CRBN mediated degradation of CK1a is also not clear.

We thank the reviewer for their additional comments. To address these insightful comments we performed the following experiments:

- a. We compared the Wnt-induced change in CK1 α half-life in a cytoplasmic cell fraction relative to the corresponding nuclear fraction. We now show that while the cytoplasmic fraction of CK1 α is degraded in response to Wnt, the nuclear fraction is not degraded. We also show that in response to Wnt the level of the PTEN associated CK1 α pool (doi: 10.1038/s41556-018-0065-8), which is cytoplasmic, does not change. Thus, our results are **not** consistent with the total pool of CK1 α being altered in response to Wnt signaling. These results and their subsequent discussion are now included in the revised manuscript (Supplementary fig. 3A-B; line 89-93).
- b. We have performed additional experiments to address the molecular mechanism of Wnt/CRBN mediated degradation of CK1 α . These results suggest that Wnt signaling induces CRBN-CK1 α association utilizing its previously described IMiD binding pocket (Fig. 4). Specifically, we now show that:
 - i. Mutation of 5 distinct residues within CRBN's IMiD binding pocket, previously shown to be essential for CRBN to associate with CK1 α in the CRBN-len-CK1 α structural model (Petzold et al. Nature, 2016), also exhibit various levels of decreased binding to CK1 α in the presence of Wnt (Fig. 4D). These CK1 α binding-deficient CRBN mutants also exhibit a reduced ability to degrade CK1 α in response to Wnt, relative to wild-type CRBN (Fig. 4C).
 - ii. In addition, besides Gly40, we mutated 3 additional amino acids in the β -hairpin loop of CK1 α that contact CRBN in the presence of lenalidomide and found that these mutations also abolish Wnt-induced CK1 α degradation (Fig. 4B).
 - iii. In collaboration with a new collaborator, who has significant experience in molecular modeling, we now provide a model suggesting that Wnt induces CRBN and CK1 α interaction via a mechanism that requires this known small molecule-binding pocket, perhaps via an endogenous small-molecule (line 270-272).

REVIEWERS' COMMENTS

Reviewer #2 (Remarks to the Author):

The authors have well addressed the issues pointed out, and gave their analysis more credibility. Therefore, I accept this paper for publication in Nature Communications.

Reviewer #3 (Remarks to the Author):

Authors have addressed my concerns.